# WHAT DOES A PLATYPUS LOOK LIKE? GENERATING CUSTOMIZED PROMPTS FOR ZERO-SHOT IMAGE CLASSIFICATION

## ABSTRACT

Open vocabulary models are a promising new paradigm for image classification. Unlike traditional classification models, open vocabulary models classify among any arbitrary set of categories specified with natural language during inference. This natural language, called "prompts", typically consists of a set of hand-written templates (e.g., "a photo of a {}") which are completed with each of the category names. This work introduces a simple method to generate higher accuracy prompts, without relying on any explicit knowledge of the task domain and with far fewer hand-constructed sentences. To achieve this, we combine open vocabulary models with large language models (LLMs) to create Customized Prompts via Language models (CuPL, pronounced "couple"). In particular, we leverage the knowledge contained in LLMs in order to generate many descriptive sentences that are customized for each object category. We find that this straightforward and general approach improves accuracy on a range of zero-shot image classification benchmarks, including over one percentage point gain on ImageNet. Finally, this simple baseline requires no additional training and remains completely zero-shot.

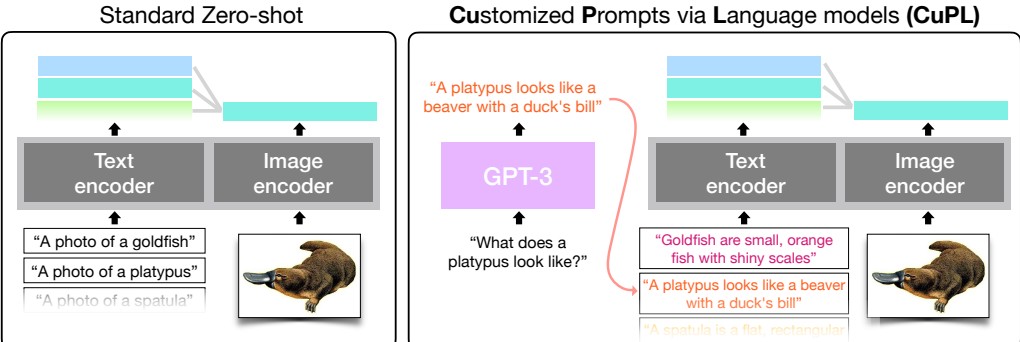

Figure 1: **Schematic of the method.** (Left) The standard method of a zero-shot open vocabulary image classification model (e.g., CLIP (Radford et al., 2021)). (Right) Our method of CuPL. First, an LLM generates descriptive captions for given class categories. Next, an open vocabulary model uses these captions as prompts for performing classification.

# 1 INTRODUCTION

Open vocabulary models (Pham et al., 2021; Jia et al., 2021; Radford et al., 2021; Yu et al., 2022a) achieve high classification accuracy across a large number of datasets without labeled training data for those tasks. To accomplish this, these models leverage the massive amounts of image-text pairs available on the internet by learning to associate the images with their correct caption, leading to greater flexibility during inference. Unlike standard models, these models classify images by providing a similarity score between an image and a caption. To perform inference, one can generate a

caption or "prompt" associated with each of the desired categories, and match each image to the best prompt. This means that categories can be selected ad hoc and adjusted without additional training.

However, this new paradigm poses a challenge:

*How can we best represent an image category through natural language prompts?*

The standard approach is to hand write a number of prompts templates (Radford et al., 2021) (e.g.,"a photo of a {}"), compile a natural language label for each category in the dataset, and create a set of prompts for each category by filling in each of these templates with the natural language labels. Then, image embeddings are matched to the nearest set of prompt embeddings and labelled with the category associated with that set of prompts (more details in Section 2).

This method has three major drawbacks. Firstly, each prompt template has to be hand-written, so having twice as many prompts for a category requires twice as much human effort. This can become costly as each new dataset typically has a different set of prompt templates (Radford et al., 2021). Secondly, the prompt templates must be general enough to apply to all image categories. For example, a prompt for the ImageNet (Deng et al., 2009) category "platypus" could only be as specific as "a photo of a {platypus}", and could not be something like "a photo of a {platypus}, a type of aquatic mammal" as that template would no longer be relevant for other image categories. Lastly, writing high performing prompt templates currently requires prior information about the contents of the dataset. For example, the list of hand-written ImageNet prompts (Radford et al., 2021) includes "a black and white photo of the {}.", "a low resolution photo of a {}.", and "a toy {}." all of which demonstrate prior knowledge about the type of representations present in the dataset. This information is not generalizable to other datasets, as ImageNet contains "black and white" and "toy" representations of its categories, but other datasets do not (e.g., FVGC Aircraft (Maji et al., 2013)).

To overcome these challenges, we propose Customized Prompts via Language models (CuPL). In this algorithm, we couple a large language model (LLM) with a zero-shot open vocabulary image classification model. We use the LLM to generate prompts for each of the image categories in a dataset. Using an LLM allows us to generate an arbitrary number of prompts with a fixed number of hand-written sentences. Additionally, these prompts are now customized to each category and can contain rich visual descriptions while still remaining zero-shot (e.g., "A platypus looks like a beaver with a duck's bill" – a sentence generated by an LLM).

We find these customized prompts outperform the hand-written templates on 15 zero-shot image classification benchmarks, including a greater than 1 percentage point gain on ImageNet (Deng et al., 2009) Top-1 accuracy and a greater than 6 percentage point gain on Describable Textures Dataset (Cimpoi et al., 2014), with fewer hand-written prompts when compared to the standard method used in Radford et al. (2021). Finally, this method requires no additional training or labeled data for either model.

## 2 METHODS

The CuPL algorithm consists of two steps: (1) generating customized prompts for each of the categories in a given dataset and (2) using these prompts to perform zero-shot image classification.

### 2.1 GENERATING CUSTOMIZED PROMPTS

This step consists of generating prompts using an LLM. For clarity, we distinguish between two different kind of prompts. The first are the prompts which cue the LLM to generate the descriptions of the dataset categories. These prompts do not describe an object, but rather prompt the description of an object (e.g., "What does a platypus look like?"). We will refer to these as "LLM-prompts".

Secondly, there are the prompts to be matched with images in the zero-shot image classification model. These are the prompts that describe a category (e.g., "A platypus looks like ..."). We call them "image-prompts." These are the output of the LLM, as examplified in Figure 2.

In this work, we use GPT-3 (Brown et al., 2020) as our LLM. To generate our image-prompts, we must first construct a number of LLM-prompt templates. While this does require some engineering

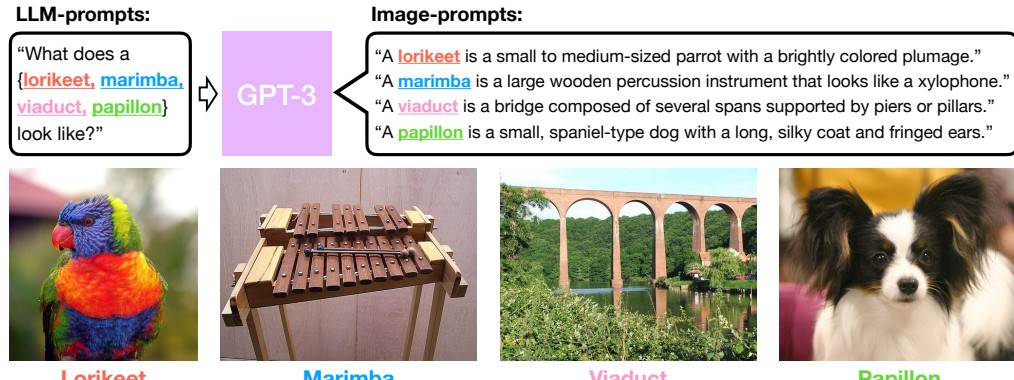

Figure 2: **Example CuPL LLM-prompts and Image-prompts.** LLM-prompts are filled in with a class name and then used as input to GPT-3, which then outputs image-prompts. Example LLM generated image-prompts and associated images from ImageNet are shown. Only image-prompts are used for the downstream image classification.

by hand, it is significantly less than the amount of hand-engineered sentences used in the standard method of creating image-prompt templates for CLIP. For example, in our ImageNet experiments, we construct 5 LLM-prompt templates compared to the 80 image-prompts used by CLIP for zero-shot ImageNet classification.

After constructing these LLM-prompts, we generate 10 different image-prompts for each of the LLM-prompts. This means for ImageNet we use an LLM to generate a total of 50 customized image-prompts for each image category. For each of these, we generate a maximum of 50 tokens, but halt a generation early if it produces a period. Additionally, we generate with a high temperature of 0.99, which encourages more diversity among the 10 generated image-prompts. We also clean each generated sentences by deleting any blank lines and adding a period at the end.

## 2.2 Utilizing Customized Prompts

After generating image-prompts for each of the categories, we then perform zero-shot image classification. While there are a number of open vocabulary models (Pham et al., 2021; Jia et al., 2021; Radford et al., 2021; Yu et al., 2022a), we report our results using CLIP (Radford et al., 2021) as this is the most popular publicly available open vocabulary model.

CLIP consists of a text encoder and and image encoder (schematic on the left side of Figure 1). In the standard setting, there are a number of hand-written templates which can be completed with the relevant category names (e.g. "A photo of a {}", "A photo of many {}"). To classify the images in a dataset, each of these templates is filled in with a given category name. Then each of these sentences is embedded via the text encoder, and all sentences completed with the same category name are averaged and normalized. This results in $n$ embeddings where $n$ is the number of categories in the dataset. Each of these $n$ embeddings is the mean of many different sentence embeddings. Then each image in the dataset is embedded using the image encoder. This embedding is compared to each of the $n$ text embeddings using cosine similarity and is labeled with the most similar one.

CuPL requires only a small adjustment from this standard practice. Instead of filling in the hand-written templates for each category, we simply replace these altogether with the sentences output by GPT-3. This means for CuPL, hand-written templates are only used as input for the LLM, while the prompts for CLIP are entirely generated text. We present 2 different setting of CuPL (as shown in Table 1), each representing a different trade-off between accuracy and hand-engineering.

**1. CuPL (base)**. This setting uses three hand-written sentence across all 15 examined datasets. We do this by constructing general LLM-prompt templates which are filled in with the category names for each dataset. Our three general templates are as follows:

Describe what a/the ___ looks like:
Describe a/the ___ :
What are the identifying characteristics of a/the ___ ?

The blank portion of this template is either filled in with the category type plus the category name (e.g. "pet" + {} for the Oxford Pets dataset (Parkhi et al., 2012) or "aircraft" + {} for FGVC Aircraft (Maji et al., 2013)) or just the category name for more general datasets like ImageNet (Deng et al., 2009). Type specification is necessary because of words that have multiple meanings. For example "boxer" from the Oxford Pets dataset can also mean a person who boxes, as opposed to a dog breed, so it is necessary to specify "Describe a pet boxer:". Similarly, "Tornado" from the FGVC Aircraft dataset can be a type of aircraft or a type of weather.

**2. CuPL (full)**. In this setting we use different LLM-prompt templates for each dataset, just as Radford et al. (2021) uses different image-prompt templates for each dataset. However, we use fewer hand-written templates overall and also contain less specific information about each dataset in the templates. For this work, each dataset has between 2 and 9 LLM-prompts which generate between 20 and 90 image-prompt per category (10 generated sentences per LLM-prompt). For ImageNet, we use the following 5 LLM-prompts: (1) "Describe what a(n) {} looks like", (2) "How can you identify a(n) {}?", (3) "What does a(n) {} look like?", (4) "A caption of an image of a(n) {}", (5) "Describe an image from the internet of a(n) {}". Example generations for each of these LLM-prompts are given for two ImageNet categories in Figure 3. Full LLM-prompts for all datasets as well as example image-prompts are given in Sections A and K of the Appendix.

# 3 EXPERIMENTS AND RESULTS

We first discuss the details of our experimental setup. We next show improvements on a wide range of image classification benchmarks. We then examine the scaling behavior with respect to the model size and report observations regarding hyperparameters such as the LLM sampling temperature. Finally, we consider and compare with other methods of obtaining descriptive captions, and provide analysis of CuPl's improvements over the standard method.

## 3.1 SETUP

Unless specified otherwise, we use CLIP with a backbone of ViT-L/14 (Dosovitskiy et al., 2020) and the GPT-3 DaVinci-002 model. Additionally, in order to perform open vocabulary image classification, each image category needs a natural language label. This is sometimes provided by the dataset, but not always (e.g. ImageNet categories are described by an id number which can map to multiple synonyms). For this work, we use the same natural language labels specified in Radford et al. (2021).

We report our findings on 15 zero-shot image recognition benchmarks: ImageNet (Deng et al., 2009), Describable Textures Dataset (DTD) (Cimpoi et al., 2014), Stanford Cars (Krause et al., 2013), Scene UNderstanding (SUN397) (Xiao et al., 2010), Food101 (Bossard et al., 2014), FGVC Aircraft (Maji et al., 2013), Oxford Pets (Parkhi et al., 2012), Caltech101 (Fei-Fei et al., 2004),

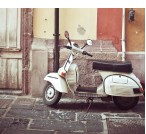
A moped typically has pedals, like a bicycle, but also has a motor.
A moped has a small engine, usually under 50cc, and generally has a maximum speed of 30mph.
A moped looks like a cross between a bicycle and a motorcycle.
The image is of a blue moped with a white seat.
A moped on a city street.

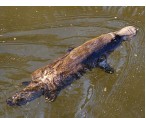
The platypus is an odd-looking animal with a duck-like bill, beaver-like tail, and otter-like body.
The easiest way to identify a platypus is by its beak, which is wide and flat like a duck's.
A platypus looks like a beaver that has a duck bill.
The image shows a platypus swimming in a body of water.
This adorable creature is a platypus, a unique mammal found in Australia.

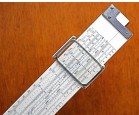
Slide rules are rectangular, and have a number of lines and markings of different lengths on them.
Slide rules have a linear scale on one edge of the rule and a logarithmic scale on the other edge.
A slide rule is a straight ruler with two scales that can slide past one another.
A slide rule is a mechanical analog computer.
This slide rule belongs to my grandfather.

Figure 3: **Example image-prompts for each of the 5 LLM-prompts**. For three ImageNet classes (moped, platypus, and slide rule), we give an example image-prompt for each of the 5 LLM-prompts used in CuPL (full) for ImageNet.

Table 1: **Performance of CuPL prompts compared to the standard, hand-written prompts in CLIP (Radford et al., 2021) on 15 zero-shot image classification benchmarks.** "Δstd" stands for the difference; green shows improvement. In addition to accuracy, we show number of prompt templates ("# hw") that are hand-written for each dataset using each method, as well as the total and unique number of hand-written templates for each method (unique number only counts templates once even if used for multiple datasets). Note that CuPL (base) uses just three hand-constructed sentence across all datasets compared to 175 in the standard method.

| | ImageNet | DTD | Stanford Cars | SUN397 | Food101 | FGVC Aircraft | Oxford Pets | Caltech101 | Flowers 102 | UCF101 | Kinetics-700 | RESISC45 | CIFAR-10 | CIFAR-100 | Birdsnap | mean | Total | Unique |
|---|---|---|---|---|---|---|---|---|---|---|---|---|---|---|---|---|---|---|
| std | 75.54 | 55.20 | 77.53 | 69.31 | 93.08 | 32.88 | 93.33 | 93.24 | 78.53 | 77.45 | 60.07 | 71.10 | 95.59 | 78.26 | 50.43 | 73.43 | | |
| # hw | 80 | 8 | 8 | 2 | 1 | 2 | 1 | 34 | 1 | 48 | 28 | 18 | 18 | 18 | 1 | | 268 | 175 |
| CuPL (base) | 76.19 | 58.90 | 76.49 | 72.74 | 93.33 | 36.69 | 93.37 | 93.45 | 78.83 | 77.74 | 60.24 | 68.96 | 95.81 | 78.47 | 51.11 | 74.15 | | |
| Δ std | +0.65 | +3.70 | -1.04 | +3.43 | +0.25 | +3.81 | +0.04 | +0.21 | +0.30 | +0.29 | +0.17 | -2.14 | +0.22 | +0.21 | +0.63 | | | |
| # hw | 3 | 3 | 3 | 3 | 3 | 3 | 3 | 3 | 3 | 3 | 3 | 3 | 3 | 3 | 3 | | 45 | 3 |
| CuPL (full) | 76.69 | 61.70 | 77.63 | 73.31 | 93.36 | 36.11 | 93.81 | 93.45 | 79.67 | 78.36 | 60.63 | 71.69 | 95.84 | 78.57 | 51.11 | 74.80 | | |
| Δ std | +1.15 | +6.50 | +0.10 | +4.00 | +0.28 | +3.23 | +0.48 | +0.21 | +1.14 | +0.91 | +0.56 | +0.59 | +0.25 | +0.31 | +0.63 | | | |
| # hw | 5 | 6 | 9 | 3 | 3 | 2 | 2 | 3 | 2 | 5 | 4 | 5 | 3 | 4 | 3 | | 59 | 45 |

Flowers 102 (Nilsback & Zisserman, 2008), UCF101 (Soomro et al., 2012), Kinetics-700 (Carreira et al., 2019), Remote Sensing Image Scene Classification (RESISC45) (Cheng et al., 2017), CIFAR-10 (Krizhevsky et al., 2009), CIFAR-100 (Krizhevsky et al., 2009), and Birdsnap (Berg et al., 2014). For the two video datasets, we extract the middle frame of the video, as is done in Radford et al. (2021).

## 3.2 RESULTS

Our results for the base prompts setting and the full prompts setting are in Table 1. We present our method's performance on 15 different image classification benchmarks, comparing both the classification accuracy and the number of hand-written sentence templates needed for each method. Note that for the standard method (Radford et al., 2021), the hand-written sentences refer to the image-prompts, while for CuPL the hand-written sentences refer to the LLM-prompts, with which image-prompts are generated.

**1. CuPL (base).** In this setting, we see performance gains in 13 out of the 15 examined datasets. Note this setting uses *just three hand-constructed sentence across all datasets*. This is in comparison to the nearly 175 unique image-prompt templates that are hand-written across all of these datasets in the standard setting. Additionally, in the standard setting these hand-constructed prompts must be very specific to the dataset (e.g., "a black and white photo of a {}.", "a plastic {}."). In comparison, CuPL (base) requires only the category type of the overall dataset and still outperforms the hand-written, domain specified baseline in almost all cases. Thus, we present this base prompt setting as a simple standard that matches or exceeds prompt engineering open vocabulary models.

**2. CuPL (full prompts).** Here we see improvements on all examined datasets. This includes large (over 1 percentage point) gains on ImageNet Top-1, DTD (texture classification), SUN397 (scene classification), FGVC Aircraft (fine-grained aircraft classification), and Flowers 102 (flower classification). While this setting requires more hand-written prompts than setting (1), it still requires significantly fewer than the baseline method (5 sentences versus 80 sentence for ImageNet), and does not include knowledge about the image domain. The full list of hand-constructed sentences for CuPL (full prompts) and the baseline method (Radford et al., 2021) can be found in Section A of the Appendix.

## 3.3 ANALYSIS AND ABLATIONS

**Model Size.** In Figure 4, we show CuPL (full prompts) at different model scales. As there are two different zero-shot models in the CuPL algorithm, we show the effects of varying each model

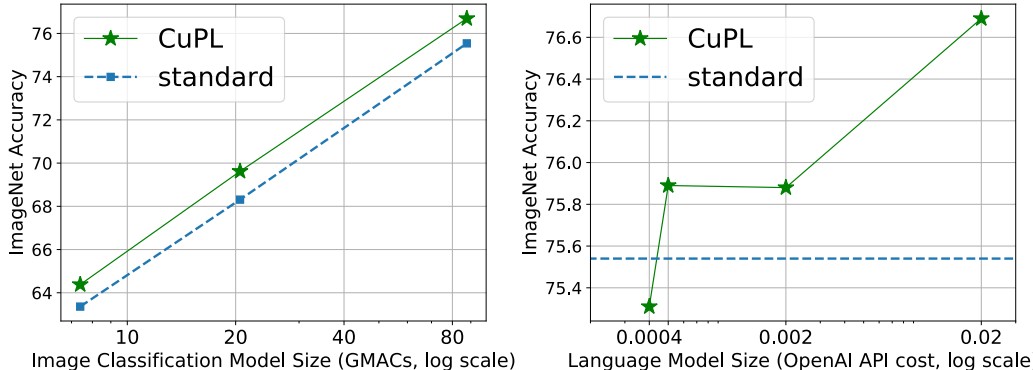

Figure 4: **Performance of CuPL as models scale.** (Left) ImageNet Top-1 accuracy for various scales of CLIP. CuPL prompts remain consistently better than standard prompts even we adjust CLIP model size (ViT-B/32, ViT-B/16, ViT-L/14). GPT-3 model set as DaVinci-002. (Right) ImageNet Top-1 accuracy for various scales of GPT-3 (ada, babbage, curie, davinci-002). Larger models produce higher accuracy. CLIP model set as ViT-L/14.

individually. On the left hand side, we vary the CLIP model used while holding the LLM constant. We see consistent gains across all model sizes. On the right hand side, we vary the size of the LLM. We plot the accuracy of the baseline as well, which does not vary as it does not utilize an LLM. We find larger models lead to higher accuracy, though the 2nd and 3rd largest models perform similarly.

**Number of Prompts.** In Figure 6, we present ablations on the number of LLM-prompts and image-prompts for CuPL (full prompts). On the left side, we show ImageNet accuracy as we increase the number of LLM-prompts. This also corresponds to the number of sentences that have to be hand-written. Notably, this methods outperforms the baseline even when using prompts generated from a single hand-written sentence. On the right hand side, we hold the number of LLM-prompts constant at 5 and adjust how many image-prompts we generate per LLM-prompt. We plot the accuracy given the total number of image-prompts (so 10 generated image-prompt per LLM-prompt corresponds to 50 total image-prompts). We see

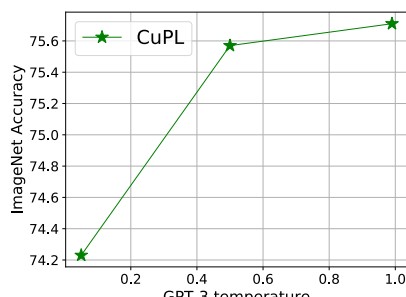

Figure 5: **Effect of LLM temperature.** More prompt diversity leads to higher performance.

that CuPL begins to outperform the baseline at just 25 image-prompts, well below the 80 image-prompts used in the baseline.

**Diversity of Prompts.** We also examine the impact of the diversity of image-prompts on ImageNet accuracy. We adjust this parameter by changing the temperature of the GPT-3 model. This value changes the likelihood of selecting lower probability tokens and makes sentences more diverse from each other. As demonstrated in Figure 5, more diverse prompts lead to higher ImageNet accuracy. Note these comparisons are done with a single LLM-prompt to save computational cost.

**WordNet Definitions and Wikipedia Descriptions.** We also consider two additional methods of obtaining descriptive sentences for each ImageNet category, other than using an LLM. Firstly, we compare CuPL (full) image-prompts with image-prompts generated using definitions of each ImageNet category. Because each ImageNet category is derived from the WordNet database (Miller, 1995), we can use the WordNet definition of each word.

We preprocess these definitions so they are of the form "A(n) {} is a ..." as not all WordNet definitions contain the name of the word itself. We also add a period to the end of each definition, as we find this increases performance. As shown in Table 2, ImageNet Top-1 accuracy with WordNet

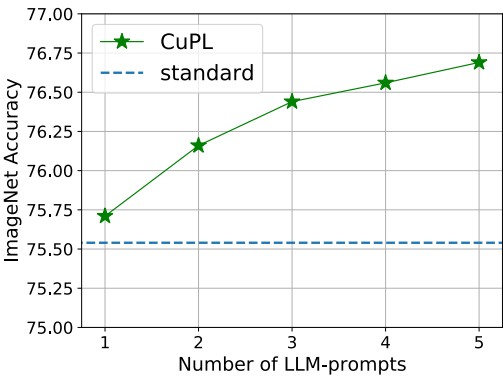 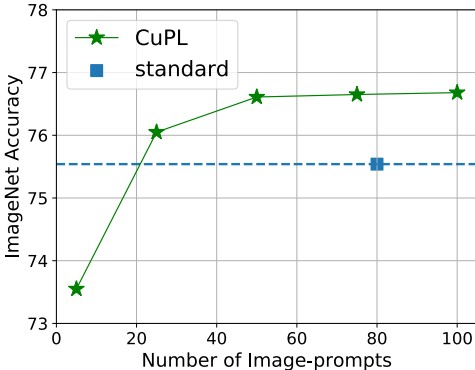

Figure 6: **Ablation on number of LLM-prompts (left) and image-prompts (right).** (Left) As number of hand-written LLM-prompts increases, so does accuracy. 10 image-prompts are generated for each LLM-prompt. Note that CuPL outperforms the baseline even with just one hand-written sentence. We add the prompts in a greedy manner, at each step adding the 10 prompts which lead to the largest performance gain. (Right) We adjust the number of image-prompts generated by a fixed number (5) of LLM-prompts. Even at 5 Image-prompts per LLM-prompt (25 prompts total), we outperform the baseline which uses 80 image-prompts.

definition prompts is below that of CuPL or standard prompts. In addition to lower accuracy, this method uses significantly more hand-constructed sentences as it requires 1000 unique hand-written definitions compared to 175 unique hand-written image-prompt templates for the standard method and 45 unique hand-written LLM-prompt templates for CuPL (full).

Secondly, we compare against prompts generated from Wikipedia articles corresponding to each Im-ageNet category, as collected in Bujwid & Sullivan (2021b). Note that the Wikipedia article does not always exactly match the natural language name of the class used by Radford et al. (2021). Additionally, 80 categories map to more than one Wikipedia article (e.g. the category associated with the natural language word "patio" is mapped to the articles for "patio" and "terrace"). In this case, we select the first associated article. We preprocess these by removing the first line (the name of the article), and then extracting the first sentence, including the final period. We find both of these preprocessing steps lead to increase in accuracy. We also truncate this sentence to the maximum allowed input length of CLIP. For the 24 ImageNet categories that do not have an associated Wikipedia page, we use the name of the category as the image-prompts. As shown in Table 2, we find Wikipedia to be less effective than standard prompts, CuPL prompts, or WordNet definitions.

Table 2: **ImageNet Top-1 accuracy for different methods of generating image-prompts.**

| Standard | CuPL | WordNet | Wiki |
|----------|------|---------|------|
| 75.54 | 76.69 | 73.44 | 68.20 |

**Ensembling with Standard Prompts.** We also consider using LLM generated prompts from CuPL (full) in addition to hand-written prompts. We do this by averaging together all the text embeddings of the CuPL prompts and hand-written prompts. As shown in Table 3, we find that for some datasets, ensembling both types of prompts outperforms CuPL prompts on their own, while for others CuPL prompts perform better. For all datasets, this ensemble performs better than standard prompts alone. However, this ensembling method requires all the hand-written effort and domain knowledge of the standard approach.

**Analysis of Accuracy Gains.** In addition to total accuracy gains, we present the per class accuracy shift between the image-prompts used in Radford et al. (2021) and CuPL, shown in Figure 7. As demonstrated, the accuracy gains seen in CuPL are not distributed uniformly through the ImageNet classes, with some classes seeing ~40 percentage point accuracy gains, and others seeing ~40 percentage point accuracy losses when compared against class accuracy with standard prompts. In other words, while CuPL sees a higher accuracy overall when compared to the standard method, the images which are correctly predicted by the standard prompts are not a subset of the images

Table 3: **Performance of the ensemble of CuPL (full) and the standard, hand-written prompts in CLIP (Radford et al., 2021)**. This ensemble outperforms the standard hand-written prompts for all examined datasets (difference shown with Δ std), and outperforms CuPL (full) for 11 datasets (difference shown with Δ CuPL)

| | ImageNet | DTD | Stanford Cars | SUN397 | Food101 | FGVC Aircraft | Oxford Pets | Caltech101 | Flowers 102 | UCF101 | Kinetics-700 | RESISC45 | CIFAR-10 | CIFAR-100 | Birdsnap | mean |
|---|---|---|---|---|---|---|---|---|---|---|---|---|---|---|---|---|
| Ensemble | 76.51 | 61.60 | 77.66 | 73.51 | 93.42 | 36.47 | 93.71 | 93.87 | 79.73 | 78.16 | 61.50 | 73.03 | 95.88 | 79.33 | 51.09 | 75.03 |
| Δ std | +0.97 | +6.40 | +0.13 | +4.20 | +0.34 | +3.59 | +0.38 | +0.63 | +1.20 | +0.71 | +1.43 | +1.93 | +0.29 | +1.07 | +0.66 | |
| Δ CuPL | -0.18 | -0.1 | +0.03 | +0.20 | +0.06 | +0.36 | -0.10 | +0.42 | +0.06 | +0.20 | +0.87 | +1.34 | +0.04 | +0.76 | -0.02 | |

which are correctly predicted by CuPL. In fact CuPL sees just over a 1 percentage point gain when compared to standard prompts, but differs in it's predictions from the standard method for 11.50% of predictions (with CuPL correct for 4.48% of these, standard correct for 3.32%, and neither correct for 3.70%).

Figure 7 also shows the classes with the 20 greatest accuracy gains and losses when comparing class accuracy with standard image-prompts (Radford et al., 2021) versus with CuPL image-prompts. Interestingly, for many of the classes which see a large accuracy gain, we see a corresponding class with a large accuracy loss that is either similar to the initial class or likely to co-occur with it (e.g. agaric/mushroom, academic gown/graduation cap, military uniform/Pickelhaube, desk/monitor).

# 4 RELATED WORK

## 4.1 NATURAL LANGUAGE DESCRIPTIONS FOR IMAGE CLASSIFICATION

Several prior works use text-based knowledge of image categories to improve classification accuracy. Elhoseiny et al. (2017) extract visual information from unstructured text descriptions collected from the internet to recognize parts of object and classify them in a zero-shot way. Reed et al. (2016) and He & Peng (2017) use natural language descriptions of bird types to train a multimodal classification model. Huang et al. (2021) use hand-collected attribute tags to attend over relevant features in images. Paz-Argaman et al. (2020) extract visual information from Wikipedia descriptions to enable zero-shot bird classification. Additional works (Shen et al., 2022; Bujwid & Sullivan, 2021a) show improvements on large datasets (e.g., ImageNet) using external information from external databases such as Imagenet-wiki and Wordnet. While these works show the effectiveness of augmenting zero-shot models with descriptive text, all of these prior works rely on external natural language databases for descriptions. This often limits the possible categories that can be classified and can require extensive preprocessing to extract visual descriptions from noisy natural language.

## 4.2 GENERATED TEXT FOR DOWNSTREAM TASKS

Recent work has utilized text generated from LLMs in a number of ways. Santurkar et al. (2022) use an LLM to paraphrase existing image captions to use as data augmentation for CLIP. Liu et al. (2022) use GPT-3 to generate knowledge on a topic when given a number of demonstrations, which is then used to improve accuracy on common sense reasoning questions. Hu et al. (2022) use a LLM to add labels to text to improve text classification accuracy. In Yu et al. (2022b), the outputs of a GPT-2 model are used to train an encoder on top of a vision model to generate multimodal image representations for a variety of tasks. Su et al. (2022) utilize a language model to perform image captioning by iteritively generating candidate image captions with a LLM and then using feedback from an open vocabulary model to align it to a given image. Similarly, Yang et al. (2022) use GPT-3 along with text descriptions of images for the Visual Question Answering (VQA) task. However, unlike CuPL these prior works are either purely language tasks (common sense reasoning, text classification) or multimodal with some language component (image captioning, VQA). In our work, we demonstrate how LLM generated text can be used to improve purely visual image classification tasks across a number of benchmarks.

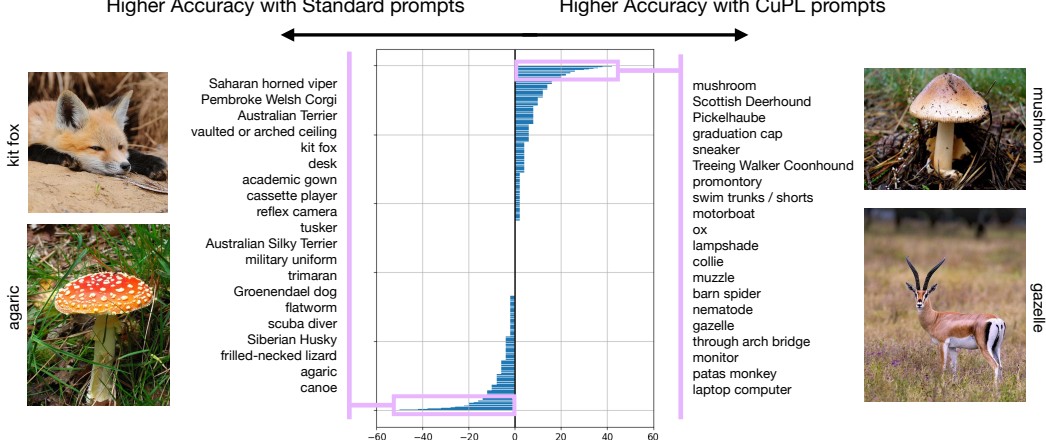

Figure 7: **Distribution of ImageNet per class accuracy difference of CuPL image-prompts versus standard image-prompts.** As shown, the accuracy gains of CuPL are not uniform across all classes. Rather, we see large gains for some classes, and losses for others. In addition, we list the classes which see the largest accuracy gains when switching to CuPL prompts (with "mushroom" having the largest gain), and the 20 classes with the largest accuracy losses (with "canoe" having the largest loss.)

## 4.3 PROMPT ENGINEERING

Previous efforts have explored methods for obtaining successful natural language prompts. For both open vocabulary image classification models as well as LLMs, the format of prompts is known to highly affect accuracy (Schick & Schütze, 2021; Radford et al., 2021; Brown et al., 2020; Gao et al., 2020). This has led to a large effort to find optimal prompt formats. Proposed methods include crowd-sourcing high performing prompts (Bach et al., 2022) as well as framing prompts to induce models to give explanations as well as answers (Wei et al., 2022; Kojima et al., 2022; Nye et al., 2021). Additional works have proposed learning prompts via gradient based methods (Zhang et al., 2021; Qin & Eisner, 2021; Li & Liang, 2021; Lester et al., 2021; Shin et al., 2020), retrieval from a database (Rubin et al., 2022), or reformatting/rephrasing existing prompts (Jiang et al., 2020; Rubin et al., 2022).

Most relevant to this work are a number of methods for designing optimal prompts for zero-shot image classification with open vocabulary models. These methods learn prompts formats which yield high accuracy for image classification using either supervised (Zhou et al., 2022; Rao et al., 2022) or unsupervised (Huang et al., 2022) methods. However, unlike these prior works this work requires no additional training or labeled data.

## 5 CONCLUSION

We demonstrate that leveraging knowledge from an LLM can immediately improve zero-shot accuracy on a variety of image classification tasks, with much less hand-engineering efforts to craft natural language prompts. Furthermore, prompts can be customized to the desired categories, rather than a general template that applies to all existing image categories. Finally, using prompts generated by LLMs lowers the barrier of prior knowledge about the dataset, which is often required when crafting prompt templates.

Querying an LLM for prompt construction is simple, straightforward and as our results suggested, immediately beneficial. The hypothesis that a joint force of LLMs and open vocabulary models would improve zero-shot image classification is thoroughly tested in this work. We hope these findings serve as a useful tool towards understanding and improving zero-shot image classification, and more generally, the consolidation of model capacities and modalities through natural language.

## 6 REPRODUCIBILITY

We have a number of measures to ensure the reproducibility of this work. First, in the supplementary material we include the code to generate image-prompts for ImageNet and evaluate the accuracy of these prompts. In Section 2, we note all hyperparameters used for the LLM. Additionally, in the appendix we include all LLM-prompts used to generate image-prompts for each of the 15 datasets. In the supplementary material, we include all generated image-prompts for all dataset, for both CuPL (base) and CuPL (full).

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

APPENDIX: *What does a platypus look like?*
*Generating customized prompts for zero-shot image classification*

OVERVIEW

A    CUPL (FULL PROMPTS) VS STANDARD PROMPTS

We detail the hand-written prompt templates used for CuPL (full prompts) versus standard CLIP Radford et al. (2021) prompt templates. For CuPL, hand-written prompt templates are needed for the LLM-prompts, while for the standard method hand-written prompt templates are needed for the image-prompts.

Note that many of the hand-written templates for the standard method encode information about the datasets. For example, "a toy {}" demonstrates knowledge that objects are sometimes represented as a toy version of an object rather than as the literal object. CuPL prompts remain much more general (e.g. "Describe what a {} looks like").

# Caltech101

| CuPL hand-written | Standard hand-written | |
|---|---|---|
| Describe what a(n) {} looks like: | a photo of a {}. | a photo of the {}. |
| Describe a(n) {}: | a painting of a {}. | a painting of the {}. |
| What are the identifying characteristics of a(n) {}? | a plastic {}. | the plastic {}. |
| | a sculpture of a {}. | a sculpture of the {}. |
| | a sketch of a {}. | a sketch of the {}. |
| | a tattoo of a {}. | a tattoo of the {}. |
| | a toy {}. | the toy {}. |
| | a rendition of a {}. | a rendition of the {}. |
| | a embroidered {}. | the embroidered {}. |
| | a cartoon {}. | the cartoon {}. |
| | a {} in a video game. | the {} in a video game. |
| | a plushie {}. | the plushie {}. |
| | a origami {}. | the origami {}. |
| | art of a {}. | art of the {}. |
| | graffiti of a {}. | graffiti of the {}. |
| | a drawing of a {}. | a drawing of the {}. |
| | a doodle of a {}. | a doodle of the {}. |

## Food101

| CuPL hand-written | Standard hand-written |
| --- | --- |
| Describe what {} looks like
Visually describe {}
How can you tell that the food in this photo is {}? | a photo of {}, a type of food. |

## Stanford Cars

| CuPL hand-written | Standard hand-written |
| --- | --- |
| How can you identify a(n) {}?
Description of a(n) {}, a type of car.
A caption of a photo of a(n) {}:
What are the primary characteristics of a(n) {}?
Description of the exterior of a(n) {}
What are the identifying characteristics of a(n) {}, a type of car?
Describe an image from the internet of a(n) {}
What does a(n) {} look like?
Describe what a(n) {}, a type of car, looks like | a photo of a {}.
a photo of the {}.
a photo of my {}.
i love my {}!
a photo of my dirty {}.
a photo of my clean {}.
a photo of my new {}.
a photo of my old {}. |

## Oxford Pets

| CuPL hand-written | Standard hand-written |
| --- | --- |
| Describe what a pet {} looks like
Visually describe a(n) '{}', a type of pet. | a photo of a , a type of pet. |

## ImageNet

| CuPL hand-written | Standard hand-written | |
| --- | --- | --- |
| Describe what a(n) {} looks like
How can you identify a(n) {}?
What does a(n) look like?
Describe an image from the internet of a(n) {}
A caption of an image of a(n) {}: | a bad photo of a {}.
a photo of many {}.
a sculpture of a {}.
a photo of the hard to see {}.
a low resolution photo of the {}.
a rendering of a {}.
graffiti of a {}.
a bad photo of the {}.
a cropped photo of the {}.
a tattoo of a {}.
the embroidered {}.
a photo of a hard to see {}.
a bright photo of a {}.
a photo of a clean {}.
a photo of a dirty {}.
a dark photo of the {}.
a drawing of a {}.
a photo of my {}.
the plastic {}.
a photo of the cool {}.
a close-up photo of a {}.
a black and white photo of the {}.
a painting of the {}.
a painting of a {}.
a pixelated photo of the {}.
a sculpture of the {}.
a bright photo of the {}.
a cropped photo of a {}.
a plastic {}.
a photo of the dirty {}.
a jpeg corrupted photo of a {}.
a blurry photo of the {}.
a photo of the {}.
a good photo of the {}.
a rendering of the {}.
a {} in a video game.
a photo of one {}.
a doodle of a {}.
a close-up photo of the {}.
a photo of a {} | the origami {}.
the {} in a video game.
a sketch of a {}.
a doodle of the {}.
a origami {}.
a low resolution photo of a {}.
the toy {}.
a rendition of the {}.
a photo of the clean {}.
a photo of a large {}.
a rendition of a {}.
a photo of a nice {}.
a photo of a weird {}.
a blurry photo of a {}.
a cartoon {}.
art of a {}.
a sketch of the {}.
a embroidered {}.
a pixelated photo of a {}.
itap of the {}.
a jpeg corrupted photo of the {}.
a good photo of a {}.
a plushie {}.
a photo of the nice {}.
a photo of the small {}.
a photo of the weird {}.
the cartoon {}.
art of the {}.
a drawing of the {}.
a photo of the large {}.
a black and white photo of a {}.
the plushie {}.
a dark photo of a {}.
itap of a {}.
graffiti of the {}.
a toy {}.
itap of my {}.
a photo of a cool {}.
a photo of a small {}.
a tattoo of the {}. |

## FGVC Aircraft

| CuPL hand-written | Standard hand-written |
|---|---|
| Describe a(n) {} aircraft | a photo of a {}, a type of aircraft. |
| Describe the {} aircraft | a photo of the {}, a type of aircraft. |

## DTD

| CuPL hand-written | Standard hand-written |
|---|---|
| What does "{}" material look like? | a photo of a {} texture. |
| What does a "{}" surface look like? | a photo of a {} pattern. |
| What does a "{}" texture look like? | a photo of a {} thing. |
| What does a "{}" object look like? | a photo of a {} object. |
| What does a "{}" thing look like? | a photo of the {} texture. |
| What does a "{}" pattern look like? | a photo of the {} pattern. |
| | a photo of the {} thing. |
| | a photo of the {} object. |

## SUN397

| CuPL hand-written | Standard hand-written |
|---|---|
| Describe what a(n) {} looks like | a photo of a {}. |
| How can you identify a(n) {}? | a photo of the {}. |
| Describe a photo of a(n) {} | |

## Kinetics-700

| CuPL hand-written | Standard hand-written |
|---|---|
| Describe the action "{}" | a photo of a person {}. |
| What does a person {} look like? | a photo of a person using {}. |
| What does the act of {} look like? | a photo of a person doing {}. |
| Describe "{}" | a photo of a person during {}. |
| | a photo of a person performing {}. |
| | a photo of a person practicing {}. |
| | a video of {}. |
| | a video of a person {}. |
| | a video of a person using {}. |
| | a video of a person doing {}. |
| | a video of a person during {}. |
| | a video of a person performing {}. |
| | a video of a person practicing {}. |
| | a example of {}. |
| | a example of a person {}. |
| | a example of a person using {}. |
| | a example of a person doing {}. |
| | a example of a person during {}. |
| | a example of a person performing {}. |
| | a example of a person practicing {}. |
| | a demonstration of {}. |
| | a demonstration of a person {}. |
| | a demonstration of a person using {}. |
| | a demonstration of a person doing {}. |
| | a demonstration of a person during {}. |
| | a demonstration of a person performing {}. |
| | a demonstration of a person practicing {}. |

## UCF 101

| CuPL hand-written | Standard hand-written |
| --- | --- |
| Describe the action of {} | a photo of a person {}. |
| What does the act of {} look like? | a video of a person {}. |
| What does a person doing {} look like? | a example of a person {}. |
| Describe "{}" | a demonstration of a person {}. |
| Describe the action "{}" | a photo of the person {}. |
|  | a video of the person {}. |
|  | a example of the person {}. |
|  | a demonstration of the person {}. |
|  | a photo of a person using {}. |
|  | a video of a person using {}. |
|  | a example of a person using {}. |
|  | a demonstration of a person using {}. |
|  | a photo of the person using {}. |
|  | a video of the person using {}. |
|  | a example of the person using {}. |
|  | a demonstration of the person using {}. |
|  | a photo of a person doing {}. |
|  | a video of a person doing {}. |
|  | a example of a person doing {}. |
|  | a demonstration of a person doing {}. |
|  | a photo of the person doing {}. |
|  | a video of the person doing {}. |
|  | a example of the person doing {}. |
|  | a demonstration of the person doing {}. |
|  | a photo of a person during {}. |
|  | a video of a person during {}. |
|  | a example of a person during {}. |
|  | a demonstration of a person during {}. |
|  | a photo of the person during {}. |
|  | a video of the person during {}. |
|  | a example of the person during {}. |
|  | a demonstration of the person during {}. |
|  | a photo of a person performing {}. |
|  | a video of a person performing {}. |
|  | a example of a person performing {}. |
|  | a demonstration of a person performing {}. |
|  | a photo of the person performing {}. |
|  | a video of the person performing {}. |
|  | a example of the person performing {}. |
|  | a demonstration of the person performing {}. |
|  | a photo of a person practicing {}. |
|  | a video of a person practicing {}. |
|  | a example of a person practicing {}. |
|  | a demonstration of a person practicing {}. |
|  | a photo of the person practicing {}. |
|  | a video of the person practicing {}. |
|  | a example of the person practicing {}. |
|  | a demonstration of the person practicing {}. |

## RESISC 45

| CuPL hand-written | Standard hand-written |
|---|---|
| Describe a satellite photo of a(n) {} 
 Describe a(n) {} as it would appear in an aerial image 
 How can you identify a(n) {} in an aerial photo? 
 Describe the satellite photo of a(n) {} 
 Describe an aerial photo of a(n) {} | satellite imagery of {}. 
 aerial imagery of {}. 
 satellite photo of {}. 
 aerial photo of {}. 
 satellite view of {}. 
 aerial view of {}. 
 satellite imagery of a {}. 
 aerial imagery of a {}. 
 satellite photo of a {}. 
 aerial photo of a {}. 
 satellite view of a {}. 
 aerial view of a {}. 
 satellite imagery of the {}. 
 aerial imagery of the {}. 
 satellite photo of the {}. 
 aerial photo of the {}. 
 satellite view of the {}. 
 aerial view of the {}. |

## Birdsnap

| CuPL hand-written | Standard hand-written |
|---|---|
| Describe what the bird {} looks like: 
 Describe the bird {}: 
 What are the identifying characteristics of the bird {}? | a photo of a {}, a type of bird. |

## Flowers 102

| CuPL hand-written | Standard hand-written |
|---|---|
| Describe how to identify a(n) {}, a type of flower 
 What does a(n) {} flower look like? | a photo of a {}, a type of flower.' |

## CIFAR-10

| CuPL hand-written | Standard hand-written |
|---|---|
| Describe what a(n) {} looks like 
 Describe a(n) {}: 
 What are the identifying characteristics of a(n) {}? | a photo of a {}. 
 a blurry photo of a {}. 
 a black and white photo of a {}. 
 a low contrast photo of a {}. 
 a high contrast photo of a {}. 
 a bad photo of a {}. 
 a good photo of a {}. 
 a photo of a small {}. 
 a photo of a big {}. 
 a photo of the {}. 
 a blurry photo of the {}. 
 a black and white photo of the {}. 
 a low contrast photo of the {}. 
 a high contrast photo of the {}. 
 a bad photo of the {}. 
 a good photo of the {}. 
 a photo of the small {}. 
 a photo of the big {}. |

# CIFAR-100

| CuPL hand-written | Standard hand-written |
|---|---|
| Describe a photo of a(n) {}: 
 What are the identifying characteristics of a(n) {}? 
 Describe what a(n) {} looks like: 
 Describe a(n) {}: | a photo of a {}. 
 a blurry photo of a {}. 
 a black and white photo of a {}. 
 a low contrast photo of a {}. 
 a high contrast photo of a {}. 
 a bad photo of a {}. 
 a good photo of a {}. 
 a photo of a small {}. 
 a photo of a big {}. 
 a photo of the {}. 
 a blurry photo of the {}. 
 a black and white photo of the {}. 
 a low contrast photo of the {}. 
 a high contrast photo of the {}. 
 a bad photo of the {}. 
 a good photo of the {}. 
 a photo of the small {}. 
 a photo of the big {}. |

## B    CuPL Base Prompts

The three general sentences used in the base prompt setting are:

Describe what a/the ___ looks like:

Describe a/the ___ :

What are the identifying characteristics of a/the ___ ?

Here we specify the type filled in for each of the examined datasets, as well as the article used for that dataset ('a(n)' or 'the'):

| Dataset | Base LLM-prompt type specification |
|---|---|
| ImageNet | a(n) {} |
| DTD | the texture {} |
| StanfordCars | the car {} |
| SUN397 | a(n) {} |
| Food 101 | the food {} |
| FGVC Aircraft | the aircraft {} |
| Oxford Pets | a pet {} |
| Caltech101 | a(n) {} |
| CIFAR-10 | a(n) {} |
| CIFAR-100 | a(n) {} |
| Flowers 102 | the flower {} |
| Kinetics-700 | the action of {} |
| UCF101 | the action of {} |
| RESISC45 | a satellite photo of {} |
| Birdsnap | the bird {} |

## C  EVALUATION METRIC

| ImageNet | DTD | Stanford Cars | SUN397 | Food101 | FGVC Aircraft | Oxford Pets | Caltech101 | Flowers 102 | UCF101 | Kinetics-700 | RESISC45 | CIFAR-10 | CIFAR-100 | Birdsnap |
|---|---|---|---|---|---|---|---|---|---|---|---|---|---|---|
| Acc. | Acc. | Acc. | Acc. | Acc. | Mean per class | Mean per class | Mean per class | Mean per class | Acc. | Mean (top1, top5) | Acc. | Acc. | Acc. | Acc. |

## D  OPEN-SOURCE LLM

While GPT-3 (Brown et al., 2020) demonstrates higher performance on a number of tasks compared to smaller open-source models, open-source models are sometime more accessible. We therefore show improvement using GPT-J-6B (Wang & Komatsuzaki, 2021), a small open-source model available on HuggingFace (Wolf et al., 2019). We find that we are able to surpass human written prompts with prompts generated by this model as shown in Table, though we still fall short of those generated by GPT-3. Additionally, we employ a number of

Table 4: **CuPL with an open-source model**. CuPL is able to improve over hand-written baselines even for smaller open-source models.

|  | ImageNet |
|---|---|
| standard | 75.54 |
| CuPL (GPT-J-6B) | 75.62 |
| CuPL (GPT-3) | 76.69 |

strategies to increase the accuracy of the lower quality GPT-J-6B generations. First, we generate at a lower temperature (0.3) to prevent irrelevant or nonsensical generations, which we find occur more frequently in smaller models. Additionally, we generate 5 times more Image-prompts per LLM-prompt than we do when using GPT-3 (Brown et al., 2020). We also add punctuation to the end of all LLM-prompts to encourage the LLM to begin new sentences. Finally, we filter out any Image-prompts which do not contain the name of the ImageNet category they are meant to describe, as well as remove a number of unicode characters from the generations (e.g. 'u2019'). We present these findings as a way to make CuPL a more accessible options until large high performance models become available to the public.

## E  SINGLE SENTENCE BASELINE

Table 5: **Single sentence baselines**. Comparison of a single hand-written Image-prompt template with a single handle written LLM-prompt as well as a single CuPL generated Image-prompt.

|  | ImageNet |
|---|---|
| a photo of a {} | 73.46 |
| CuPL (1 hand-written) | 75.71 |
| CuPL (1 generated) | 74.24 |

One of the primary benefits of CuPL is that it decreases the amount of necessary hand-engineering. However, this could also be done by decreasing the number of total hand-written templates used, which comes at a loss in performance. We present this as a 'low effort' baseline, where we use only the hand constructed template of 'a photo of a {}'. We compare this with two CuPL baselines. The first is the baseline in which we also only construct one hand written template: 'Describe what a {} looks like'. We then use this to generate 10 Image-prompts. The second baseline is a single CuPL generated sentence, generated with the prompt 'Describe what a {} looks like'. For this experiment, we generate at a temperature of 0.3 as we find that higher tempuratures are only helpful when we are able to ensemble many diverse prompts, not when we are limited to one. We find that CuPL outperforms a single hand-written template under both of these settings, as shown in Table 5.

# F ROBUSTNESS

In addition to the previously mentioned benefits of open vocabulary models, one of the important advances made by CLIP (Radford et al., 2021) is an increased robustness on out-of-distribution data. Fine-tuning has been shown to degrade performance on out-of-distribution tasks (Wortsman et al., 2022), however zero-shot CLIP is robust to these distribution shifts. We show improvement on two common distribution shifts in Table 6, demonstrating that CuPL maintains the robustness of CLIP.

Table 6: **Robustness of CuPL.** CuPL accuracy on two common ImageNet variants, using CuPL ImageNet Image-prompts. CuPL improves performance on both of these variants, demonstrating that CuPL improves accuracy on in-distribution tasks, while maintaining robustness to distribution shifts.

|  | ImageNet (Deng et al., 2009) | ImageNet-V2 (Recht et al., 2019) | ImageNet-Sketch (Wang et al., 2019) |
|---|---|---|---|
| Standard | 75.54 | 69.86 | 59.60 |
| CuPL | 76.69 | 70.85 | 60.05 |

# G CuPL IMPROVEMENT ANALYSIS

## G.1 VISUAL SIMILARITY ANALYSIS

In Figure 7, we provide initial analysis on the categories where the CuPL algorithm improves the most over standard hand-written prompts. We find that the improvement is not uniformly distributed, but rather some classes see a large improvement, while others see a decrease in per class accuracy. Interestingly, there are often two similar categories where one sees a large increase in accuracy and the other sees a decrease. For example, the 'mushroom' class has an approximately 40pp increase, while 'agaric' (a subclass of mushroom) is one of the classes with the largest drop in accuracy.

In order to better understand this phenomenon, we examine the change in accuracy between CuPL and the standard method of prompting in the image embedding space. Thus we are able to visualize the close relationship between categories like 'agaric' and 'mushroom'. In order to be able to visualize the high dimensional CLIP image embedding in two dimensions, we utilize the t-distributed stochastic neighbor embedding algorithm (Van der Maaten & Hinton, 2008). Figure 8 visualizes image features (reduced into two dimensions) in relation to CuPL improvement.

As was suggested by Figure 7, we see in Figure 8 that when there is a class that has a large increase in accuracy with CuPL prompts ('mushroom', 'graduation cap', 'monitor') there is often a decrease in class accuracy for a visually related class. This means that when choosing between two similar or co-occuring classes, CuPL has a different distribution of classification than the standard method (e.g. the standard method prefers 'canoe' over 'paddle' much more strongly than CuPL). This suggests that the overall accuracy improvement of CuPL over the standard method may come (at least in part) from better distinguishing between two visually similar classes. While it may be over-correcting from the mistakes of the standard method (as demonstrated by the drop in accuracy in one of the two similar classes), the CuPL predictions appear to be overall more accurate, as demonstrated by the overall higher accuracy.

## G.2 CO-OCCURRENCES BETWEEN OBJECTS

Many of the frequently confused pairs in Figure 8 are objects that are likely to occur in an image (i.e. 'canoe'-'paddle' or 'graduation cap'-'academic gown' or 'monitor'-'desk'). One potential benefit of CuPL captions is that they are able to capture co-occurrences as well. For example, one CuPL prompt for the 'canoe' class is *A canoe is typically a narrow boat with pointed ends that is propelled with a paddle.* This caption contains the word 'paddle' which is frequently confused with 'canoe' and likely to be present in images, even where the correct label is 'canoe'. We therefore investigate the effectiveness of CuPL captions on images which contain more than one ImageNet object.

We attain this by using the ImageNet-ReaL dataset (Beyer et al., 2020) which relabels ImageNet images with all applicable labels, so an image with both a 'canoe' and a 'paddle' would have both

Figure 8: **Visualizing of image embedding of ImageNet classes compared to CuPL improvement on that class.** Each point on this figure represents the average image embedding of an ImageNet class, which has been reduced to two dimentions using t-sne (Van der Maaten & Hinton, 2008). We see that when there is a class with a large improvement compared to the baseline, it is often visually similar to a class which has a decrease in accuracy. This suggests that CuPL's improved accuracy may be due in part to an increased ability to distinguish similar classes compared to the baseline.

labels. We then tag images as having multiple ImageNet objects or only one ImageNet object based on the ReaL dataset. Finally, we compute ImageNet accuracy across each of these two sets (using standard ImageNet labels). Results are given in Table 7.

Table 7: **Standard versus CuPL accuracy based on number of ImageNet classes present in image.** We use the ImageNet-ReaL dataset (Beyer et al., 2020) to find images which have more than one applicable ImageNet label. We then present the accuracy for standard prompts and CuPL prompts using standard ImageNet labels, split by images which contain only one possible ImageNet class and images which may contain multiple classes.

|  | One class present (85.1% of ims) | Multiple classes present (14.9% of ims) |
| --- | --- | --- |
| Standard | 79.89 | 51.59 |
| CuPL | 80.79 | 53.58 |

## H   ERROR ANALYSIS

In Figure 9, we present an error analysis of our model using two different metrics. The first is an analysis between the model prediction and the correct class using the visual similarity of these labels. To capture this, we first attain an average visual embedding of each class by taking the mean of each image in that class and then normalizing that mean. Then for each class we rank how similar each of the other 999 classes are by the distance between these embeddings. If the models makes an incorrect prediction, but it predicts the class with the closest embedding to the correct label, then we refer to this as an *image offset* of 1.

Additionally, we examine the prediction errors in terms of the linguistic similarity of the labels. We do this with the WordNet (Miller, 1995) similarity of two labels. For example, if the label of the prediction and the ground-truth label share the same parent in the WordNet tree, that is a *WordNet offset* of 2.

While slight, there is a difference in the errors made by CuPL compared to the errors made by the baseline as shown in Figure 9. CuPL is more likely to have an error that has an image offset of 1

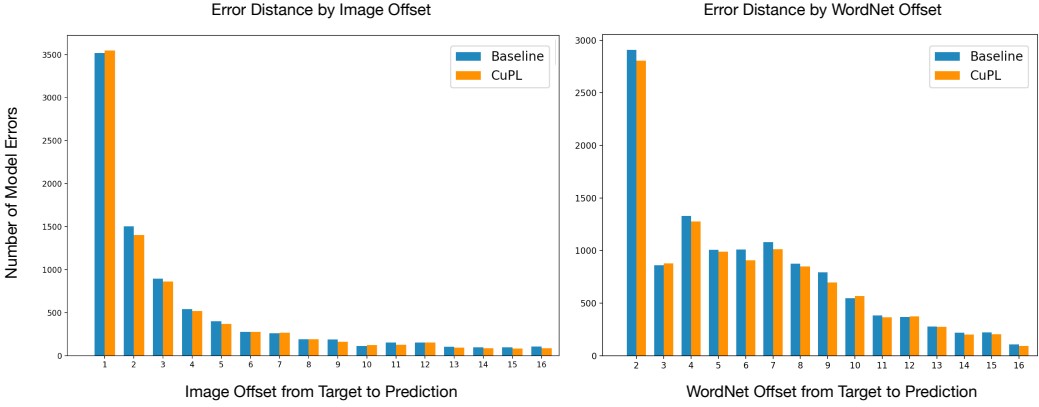

Figure 9: **Error analysis of CuPL and baseline comparing models errors visually and linguistically to ground truth labels.** We compare the errors made by CuPL to the errors made by the baseline methods. We find that the errors made by CuPL are more likely to be the most visually similar class to the ground truth label when compared to the errors made by the baseline. However, the errors made by the baseline are more likely to be the most linguistically similar class according to the WordNet (Miller, 1995) heirarchy. This suggests that visual information is being extracted from the CuPL descriptions to make visually consistent predictions.

than the baseline. However, the baseline is more likely to have an error that has a WordNet offset of 1 than CuPL. This implies that CuPL may be taking advantage of the visually descriptive language of the captions, as even when the model makes errors, they tend to favor categories that are visually similar to the ground truth. However, the baseline method does not have visual descriptions in its Image-prompts which may lead to its errors aligning more linguistically with the ground-truth.

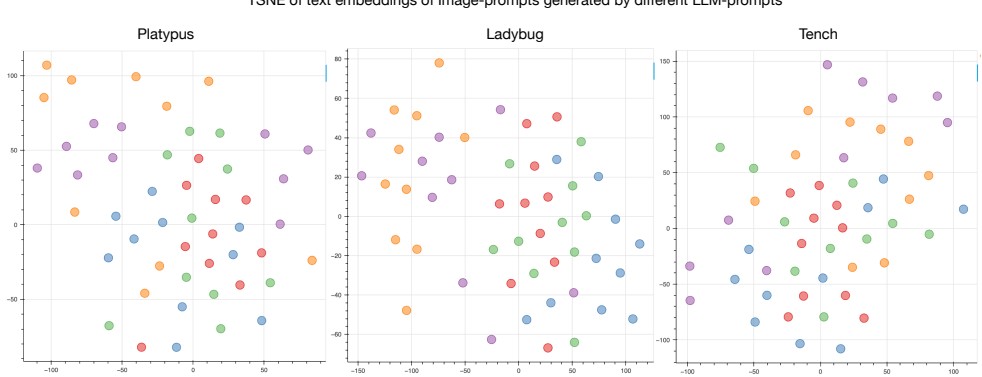

Figure 10: **Visualization of embeddings of Image-prompts generated with various LLM-prompts.** Each point represents the image embedding of one Image-prompts for the stated category with has been reduced to two dimensions using t-sne (Van der Maaten & Hinton, 2008). Image-prompts with the same color are generated by the same LLM-prompts.

## I  IMAGE-PROMPT DISTRIBUTION

When generating Image-prompts, we use an ensemble of prompts generated by different LLM-prompts (e.g. 'What does a {} look like?). As we find that the diversity of Image-prompts is correlated with accuracy, it is valuable to understand how different LLM-prompts affect the diversity of Image-prompts. To accomplish this, we visualize the text embedding of Image-prompts for a selection of classes using t-sne (Van der Maaten & Hinton, 2008) dimension reduction. We then color Image-prompts that were generated by the same LLM-prompt. As shown in Figure 10, we

find that while there is a slight clustering of Image-prompts by LLM-prompts, there is also a large amount of overlap.

## J TEMPERATURE ANALYSIS

We provide several further analyses of the effect of temperature in prompt generation. First, we visualize the distribution of Image-prompts that have been generated with a variety of different temperatures. We do this by selecting prompts for 3 different ImageNet classes at 3 different temperatures. We then perform dimentionality reduction on the text embeddings of these prompts in order to visualize their distribution. As shown in Figure 11, Image-prompts generated with a temperature of 0.1 are clustered in a few different locations. Image-prompts generated with a temperature of 0.5 are more widely, and Image-prompts generated with a temperature of 0.9 have a similar, but even wider distribution.

Additionally in Section K, we give all generated prompts for the ImageNet class 'Tench' at three different temperatures. At the lowest temperature, the generated Image-prompts are nearly identical when generated with the same LLM-prompts. As the temperature increases, so does the difference in the generated prompts.

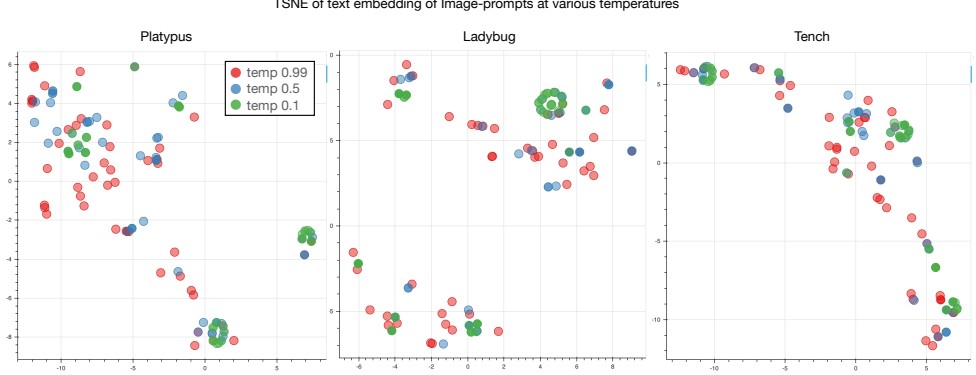

Figure 11: **Visualization of embeddings of Image-prompts generated with various temperatures.** Each point represents the image embedding of one Image-prompts for the stated category with has been reduced to two dimensions using t-sne (Van der Maaten & Hinton, 2008). Prompts generated with a higher temperature cover a wider distribution.

## K EXAMPLE GENERATED IMAGE-PROMPTS

A selection of LLM-generated image-prompts for a subset of ImageNet categories. We give all 50 image-prompts for the first ImageNet category of "Tench" and then 10 randomly selected prompts for a number of randomly selected ImageNet categories.

### K.1 ALL GENERATED IMAGE-PROMPTS FOR "TENCH" CATEGORY

**Temperature = 0.99**

```
"A tench is a freshwater fish of the carp family.",
"A tench is a freshwater fish that is typically brown or olive in
color.",
"A tench is a fresh water fish that can grow up to 2 feet in length.",
"A tench is a freshwater fish of the family Cyprinidae.",
"A tench is a freshwater fish of the carp family.",
"A tench is a freshwater fish with a dark green back and light-colored
sides.",
"Tench are a freshwater fish found in Europe.",
"A tench is a small freshwater fish in the carp family.",
```

"A tench is a heavyset freshwater fish with a mottled brown body and a
small, flat head.",
"A tench is a freshwater fish that looks similar to a carp.",
"A tench is a freshwater fish in the carp family.",
"A tench is a freshwater fish of the Cyprinidae family.",
"The tench is a freshwater fish of the Cyprinidae family.",
"The tench is a fresh-water fish in the family Cyprinidae.",
"The easiest way to identify a tench is by its herringbone-patterned
scales.",
"A tench is a freshwater fish of the carp family.",
"Tench are a freshwater fish found in Europe.",
"Tench have a large, slimy body with scales that have a green hue.",
"The tench is a freshwater fish belonging to the carp family.",
"A tench is a freshwater fish of the Cynoglossidae family.",
"A tench is a freshwater fish in the carp family.",
"Tensch are freshwater fish with Olive Green backs, shading to Yellowish
on the sides.",
"A tench looks like a green freshwater fish with a brownish hue.",
"A tench looks like a freshwater fish with a dark olive-green back,
fading to yellowish-brown on the sides.",
"A tench usually has olive-green skin with dark spots, and a
orange-yellow underbelly.",
"Tench are a freshwater fish that can grow up to 70cm long! They have
olive-brown skin with dark spots, and their meat is white and firm.",
"A tench is a freshwater fish with a sturdy body and a greenish-brown
coloration.",
"A tench is a freshwater fish that can grow up to about two feet long.",
"A tench is a freshwater fish in the carp family.",
"A tench is a large, freshwater fish with a thick body and large head.",
"The image is of a tench fish swimming in water.",
"The image is of a tench fish swimming in a pond.",
"The tench is a freshwater fish native to Europe.",
"This image shows a large, dark green tench swimming in a pond.",
"An image of a tench from the internet would likely show a dark green
fish with a lighter underside.",
"The image is of a tench fish.",
"The image is of a tench fish on a white background.",
"A tench is a freshwater fish of the Cyprinidae family.",
"The image is of a tench swimming in a murky pond.",
"In the image, a tench swims in a pond with lily pads.",
" A tench in a river.",
"A tench (Tinca tinca) is a freshwater fish in the carp family that is
found throughout Europe.",
" Tench (Tinca tinca), a member of the carp family (Cyprinidae), native
to Eurasia.",
" A tench, a freshwater fish in the family Cyprinidae.",
" The tench (Tinca tinca) is a freshwater fish of the cyprinid family
found throughout Eurasia.",
" A tench in a Finnish lake.",
"A tench (Tinca tinca) is a freshwater fish belonging to the carp family
(Cyprinidae).",
"A tench in a fishpond.",
" The common tench is a freshwater fish of the cyprinid family found
throughout Eurasia.",
"Tench (Tinca tinca) in a pond."

## Temperature = 0.5

"A tench is a freshwater fish that is typically greenish-brown in color
with a brassy sheen.",
"A tench is a freshwater fish that typically has a dark green back, light
brown sides, and a white belly.",
"A tench is a freshwater fish that can grow up to two feet long.",
"Tench are a freshwater fish found in Europe.",

"A tench is a freshwater fish that is typically olive green in color with
dark spots.",
"A tench is a freshwater fish that is typically olive green in color with
a brownish tint.",
"A tench is a freshwater fish of the Cyprinidae family.",
"Tench are a freshwater fish found in Europe.",
"A tench is a freshwater fish that has a dark green back, light brown
sides, and a white belly.",
"A tench is a freshwater fish that is typically olive-green in color with
a brownish dorsal fin.",
"A tench is a freshwater fish of the cyprinid family.",
"A tench is a freshwater fish of the carp family.",
"A tench is a freshwater fish that is typically olive green in color with
a brownish back.",
"The tench is a freshwater fish of the carp family Cyprinidae.",
"A tench is a freshwater fish that is typically greenish-brown in
color.",
"A tench is a freshwater fish of the carp family.",
"A tench is a freshwater fish of the carp family.",
"Tench have olive green backs and flanks, with yellowish bellies.",
"A tench is a freshwater fish of the carp family.",
"A tench is a freshwater fish of the cyprinid family.",
"A tench is a freshwater fish that can grow up to 30 inches long.",
"A tench is a freshwater fish that is typically greenish-brown in
color.",
"A tench is a freshwater fish that can grow up to about two feet long.",
"A tench is a freshwater fish with a brownish-green back and sides, and a
yellowish-brown belly.",
"A tench is a freshwater fish that can grow up to two feet long.",
"A tench is a freshwater fish that looks similar to a carp.",
"A tench is a freshwater fish that is typically greenish-brown in
color.",
"A tench is a freshwater fish that is part of the carp family.",
"A tench is a freshwater fish of the cyprinid family.",
"A tench is a freshwater fish that can grow up to two feet long.",
"The image is of a tench fish swimming in water.",
"The image is of a tench fish swimming in a pond.",
"The image is of a tench fish swimming in a pond.",
"The image is of a tench fish swimming in a pond.",
"The image is of a tench fish swimming in a pond.",
"In the image, a tench is swimming in a pond with lily pads.",
"The image is of a tench fish swimming in a pond.",
"The image is of a tench fish swimming in a pond.",
"The image is of a tench fish swimming in a pond.",
"The image is of a tench fish swimming in a pond.",
"A tench fish, native to Europe, characterized by its greenish-brown
color and spots.",
" A tench (Tinca tinca) in a pond.",
"A tench (Tinca tinca) is a freshwater fish belonging to the carp family
(Cyprinidae).",
"A tench (Tinca tinca) is a freshwater fish in the carp family.",
"A tench (Tinca tinca) is a freshwater fish in the carp family
(Cyprinidae).",
" A tench in a river.",
"A tench (Tinca tinca) is a freshwater fish in the carp family
(Cyprinidae).",
"A tench (Tinca tinca) in a pond.",
" A tench (Tinca tinca) in a garden pond.",
" A tench in a river."

## Temperature = 0.1

"A tench is a freshwater fish that is typically olive green in color with
dark spots.",

```
"A tench is a freshwater fish that can grow up to 30 inches long.",
"A tench is a freshwater fish that can grow to a length of over two
feet.",
"A tench is a freshwater fish that can grow up to two feet long.",
"A tench is a freshwater fish that can grow up to two feet long.",
"A tench is a freshwater fish that is typically olive green in color with
dark spots.",
"A tench is a freshwater fish that can grow up to two feet long.",
"A tench is a freshwater fish that can grow up to two feet long.",
"A tench is a freshwater fish that can grow up to two feet long.",
"A tench is a freshwater fish that is typically olive green in color with
dark spots.",
"A tench is a freshwater fish of the carp family.",
"A tench is a freshwater fish of the carp family.",
"A tench is a freshwater fish of the carp family.",
"A tench is a freshwater fish of the carp family.",
"A tench is a freshwater fish of the carp family.",
"A tench is a freshwater fish of the carp family.",
"Tench have a dark green back, light olive sides, and a yellowish
belly.",
"A tench is a freshwater fish of the carp family.",
"A tench is a freshwater fish of the carp family.",
"A tench is a freshwater fish of the carp family.",
"A tench is a freshwater fish that can grow up to two feet long.",
"A tench is a freshwater fish that can grow up to two feet long.",
"A tench is a freshwater fish that can grow up to two feet long.",
"A tench is a freshwater fish that can grow up to two feet long.",
"A tench is a freshwater fish that is typically olive green in color with
a brownish dorsal side.",
"A tench is a freshwater fish that can grow up to two feet long.",
"A tench is a freshwater fish that can grow up to two feet long.",
"A tench is a freshwater fish that is typically olive green in color with
a brownish dorsal side.",
"A tench is a freshwater fish that can grow up to two feet long.",
"A tench is a freshwater fish that can grow up to two feet long.",
"The image is of a tench fish swimming in a pond.",
"The image is of a tench fish swimming in a pond.",
"The image is of a tench fish swimming in a pond.",
"The image is of a tench fish swimming in a pond.",
"The image is of a tench fish swimming in a pond.",
"The image is of a tench fish swimming in a pond.",
"The image is of a tench fish swimming in a pond.",
"The image is of a tench fish swimming in a pond.",
"The image is of a tench fish swimming in a pond.",
"The image is of a tench fish swimming in a pond.",
"A tench (Tinca tinca) is a freshwater fish in the carp family.",
"A tench (Tinca tinca) is a freshwater fish of the carp family
(Cyprinidae).",
"A tench (Tinca tinca) is a freshwater fish in the carp family.",
"A tench (Tinca tinca) is a freshwater fish in the carp family.",
"A tench (Tinca tinca) is a freshwater fish in the carp family.",
"A tench (Tinca tinca) is a freshwater fish in the carp family.",
"A tench (Tinca tinca) is a freshwater fish in the carp family.",
" A tench (Tinca tinca) in a garden pond.",
" A tench (Tinca tinca) in a garden pond.",
"A tench (Tinca tinca) is a freshwater fish of the carp family
(Cyprinidae)."
```

## K.2 SAMPLE GENERATED IMAGE-PROMPTS FOR RANDOMLY SELECTED IMAGENET CATEGORIES

```
"bubble":
"A bubble is a sustained period of inflated asset prices.",
```

"A bubble looks like a sphere of air.",
"A bubble is often characterized by rapidly increasing prices in an asset
or security, followed by a sharp decrease in prices.",
"A bubble looks like a small, round, thin film of soap filled with air.",
"A bubble looks like a round sphere of soap film.",
" \"A bubble being blown in the park.",
"A close-up of a soap bubble with a thin film of water in between two
layers of air.",
"A bubble is a spherical shape made up of a thin film of soap water.",
"A bubble is a circle of air surrounded by water.",
"A bubble looks like a round, thin layer of soap surrounding a pocket of
air."

"kit fox":
"A kit fox is a small species of fox, about the size of a domestic cat.",
"You can identify a kit fox by its small size, its big ears, and its
long, bushy tail.",
" A kit fox laying on the ground in a desert habitat.",
"A kit fox is a small fox found in North America.",
"A kit fox is a small fox with a sleek coat of fur.",
"A kit fox is a small fox with large ears, a long, black-tipped tail, and
pale fur.",
"A kit fox has a reddish coat, with white patches on its chest and
throat.",
"This kit fox has a reddish coat and large ears.",
"A kit fox looks like a small fox with a pointed nose, large ears, and a
long, bushy tail.",
"A kit fox is a small species of fox."

"toy terrier":
"A toy terrier looks like a very small version of a terrier.",
"The image is of a toy terrier that is mostly white with brown spots.",
"You can identify a toy terrier by looking for a compact, short-legged
dog with a short muzzle.",
" Cute little guy.",
"Toy terriers are miniature versions of terriers, such as the Jack
Russell Terrier.",
"A toy terrier is a small, lightweight breed of dog.",
"The image is of a small, brown toy terrier.",
"A toy terrier typically has a long, narrow head with pointy ears, and a
small, compact body.",
"A toy terrier is a small, short-legged dog with a long body, pointy
nose, and large ears.",
"A small, brown and white toy terrier is sitting on a beige couch,
looking at the camera."

"mousetrap":
"In the image, there is a mousetrap made of wood and metal.",
"An image of a mousetrap from the internet would most likely show a
traditional wooden mousetrap with a metal spring.",
"A mousetrap is a device made to catch and kill mice.",
"A mousetrap is a small device that is used to catch mice.",
"The classic mousetrap consists of a wooden base with a metal spring
mounted on one end.",
"The mousetrap is a small wooden box with a metal spring inside.",
"The mousetrap is a simple device that has been used for centuries to
catch mice.",
"The classic mousetrap – simple, effective, and deadly.",
"The most common way to identify a mousetrap is by its small size and
rectangular shape.",
"A mousetrap typically has a wire or wooden frame that is baited with
food and springs open quickly to snap shut on the mouse when it attempts
to steal the bait."

"dog sled":

"A dog sled is a vehicle on runners, typically with a thin frame and
a flat bottom, that is used to convey goods or passengers over snow or
ice.",
"A dog sled is a toboggan pulled by dogs, typically over snow.",
"A dog sled looks like a bed on runners that is pulled by dogs.",
"A dog sled is typically a heavy frame on runners that is pulled by one
or more dogs.",
"A dog sled is traditionally a sled pulled by dogs, used for
transportation, racing, or other purposes.",
" teamA dog sled team is a group of dogs that are harnessed together to
pull a sled.",
"A dog sled looks like a small, open vehicle that is pulled by one or
more dogs.",
"Dog sledding in winter.",
" pulled by huskiesThe image is of a dog sled pulled by huskies.",
"A dog sled looks like a large cart that is pulled by a team of dogs."

"geyser":
"A geyser is a hot spring that periodically erupts, shooting a column of
water and steam into the air.",
"A geyser looks like a hole in the ground that sometimes spurts hot water
and steam into the air.",
"Geysers are hot springs that periodically spout water and steam into the
air.",
"The image is of a geyser erupting.",
"A geyser is a hot spring that periodically erupts, spraying water into
the air.",
"A geyser typically looks like a cone of rocks with a small hole at the
top.",
"A geyser is a hot spring where water intermittently boils, sending a jet
of hot water and steam into the air.",
"A geyser is a hot spring that periodically shoots a stream of hot water
and steam into the air.",
"The image is of a geyser shooting water high into the air.",
"A geyser looks like a column of water that shoots into the air and then
falls back down."

"Schipperke":
"A Schipperke is a small, Belgian breed of dog.",
"A Schipperke is a small black Belgian dog with a rat-like tail.",
"The image is of a black and white dog with pointy ears and a long
body.",
"A Schipperke is a small, black, Belgian breed of dog.",
"A Schipperke is a small, black, Belgian breed of dog that closely
resembles a fox.",
"A Schipperke is a small Belgian breed of dog that resembles a fox.",
"Schipperkes have a long, black coat and a pointed muzzle.",
"A Schipperke is a small dog breed with a fox-like appearance.",
"It's a photo of a black and tan Schipperke dog standing in front of a
brick wall.",
"Black, small, spitz-type dog with a long, fox-like snout, large erect
ears, and a long, high-set tail."

"go-kart":
"A go-kart typically looks like a small car or buggy with a small engine
in the back.",
"A go-kart is a small vehicle with four wheels, a steering wheel, and a
gas pedal.",
"A go-kart is a small vehicle with a steering wheel, pedals, and an
engine.",
"Two young girls in go-karts race down a path in a park.",
"A go-kart is a small, lightweight vehicle with four wheels and a simple,
open frame.",
"A go-kart is a small, open-wheeled vehicle used for racing.",
"Two kids racing go-karts on a dirt track.",

"A go-kart is a small, racing car.",
"A go-kart typically looks like a small, open-wheeled car.",
"A go-kart is a small, lightweight vehicle with four wheels that is
propelled by a small engine."

"black-and-white colobus":
"The Black-and-White Colobus is a type of Old World monkey, found in
Africa.",
"A black-and-white colobus has long black fur, and a white face with a
black triangle around the eyes.",
"Colobus Monkey in the TreesThis elegant colobus monkey is swinging
through the trees in search of food.",
" monkeyIn this image, a black-and-white colobus monkey is shown perched
atop a tree branch.",
"The black-and-white colobus monkey is one of the most beautiful and
distinctive of all the colobus monkeys.",
"The black-and-white colobus monkey is a species of primate in the
Colobidae family.",
"Colobus monkeys are generally black with white patches on their face,
back, and sides.",
"The black-and-white colobus is a species of Old World monkey.",
" monkeyThe image is of a black-and-white colobus monkey sitting on a
tree branch.",
" monkeyIn the image, the black-and-white colobus monkey is sitting in a
tree."

"sock":
"A sock usually has a cuff at the top, and a heel at the bottom.",
" A black sock with a white line running down the middle.",
"A sock is typically a garment worn on the feet and made from a soft
material, such as cotton.",
"A sock normally has a heel, toe and a cuff at the top.",
"A sock is a small amount of money that is given to someone without them
knowing.",
"A sock is an article of clothing worn on the feet.",
"A sock is a piece of clothing that is worn on the feet.",
"A sock is a tubular garment that covers the foot and ankle.",
"This image is of a blue and white striped sock.",
"There are many ways that you can identify a sock."

"Cocker Spaniel":
"A Cocker Spaniel is a medium sized dog with long, floppy ears, and a
silky coat that is usually either brown or black.",
"The Cocker Spaniel has a long floppy ears, a silky coat, and a bushy
tail.",
"A Cocker Spaniel has a long, black muzzle and big, brown eyes.",
"The Cocker Spaniel is a breed of dog.",
"An image of a Cocker Spaniel from the internet shows a small brown and
white dog with long floppy ears.",
"The image is of a Cocker Spaniel with short, brown fur and long, floppy
ears.",
"Cocker spaniels have long, floppy ears and a long, silky coat.",
"A Cocker Spaniel is a small to medium sized dog.",
"The image is of a light brown and white Cocker Spaniel standing on a
green grassy field with its head turned to the side.",
"A Cocker Spaniel has a long, silky coat that is usually either black,
brown, or golden."

"southern black widow":
"A southern black widow spider perched atop a web.",
"Female southern black widows have a black body with a red hourglass
shape on their abdomen.",
"The southern black widow is black with a red hourglass shape on its
belly.",

"Female southern black widow spiders are black with a characteristic red hourglass-shaped mark on their ventral abdomen.",
" spiderThe image is of a large, black spider with a red hourglass shape on its abdomen.",
"A southern black widow is a spider that is black with a red hourglass shape on its abdomen.",
"A southern black widow spider is a small, black spider with a red hourglass-shaped mark on its underside.",
"There's an image on the internet of a southern black widow that's really cool.",
"A southern black widow can be identified by its black coloration with a red hourglass shape on its abdomen.",
"A Southern black widow is a type of spider that is black with a red hourglass shape on its belly."

"catamaran":
"A catamaran is a sailboat that has two hulls, or wide bodies, that are connected by beams.",
"The easiest way to identify a catamaran is by its two hulls.",
"A catamaran is a type of boat that has two parallel hulls.",
"A catamaran is a multi-hulled vessel with two parallel hulls of equal size.",
"My dream boat! A sleek catamaran that can zip through the waves.",
"A catamaran is a multi-hulled vessel with two parallel hulls of equal size.",
"The image is of a white catamaran with blue trim.",
"A catamaran is a type of boat that has two hulls, or platforms, that are parallel to each other.",
"The image is of a yellow catamaran with white trim, sitting in calm water.",
"A catamaran is a type of sailing vessel that consists of two parallel hulls of equal size."

"beach":
".",
"Blue skies, white sands, and clear turquoise waters make this beach a paradise.",
"A beach usually has sand and water.",
"Beach identification can be accomplished through the identification of physical characteristics.",
"A beach is a naturally occurring feature of the landscape.",
"The sun sets over the ocean, casting a beautiful orange hue in the sky.",
"A beach is a large body of water with sand or small rocks at the shore.",
"In the image, the beach is Brilliant white with crystal blue waters.",
"The beach looks like a long strip of land next to the ocean.",
"A beach typically looks like a large, flat expanse of sand with some rocks or other natural features nearby."

"rotary dial telephone":
"A rotary dial is a device used to dial telephone numbers.",
"Rotary dial telephone from the mid-20th century.",
"A rotary dial telephone is a phone with a circular dial on the front face.",
"When looking at a rotary telephone, you can tell it is a rotary phone by the Place the phone's receiver on your ear and listen for a dial tone.",
"A rotary dial phone is an older model phone that has a circular device with numbers on it that you rotate with your finger to dial a number.",
"A rotary dial telephone is a type of telephone that uses a mechanical dial to select the telephone number that a user wishes to call.",
"History of the Rotary Dial Telephone.",
"A rotary dial telephone is an old-fashioned telephone that has a round dial on the front of it.",

```
" rotary dial telephone looks like a classic telephone with a rotary dial
on the base unit.",
"The image is of a rotary dial telephone on a black background."
```

