# OpenReview forum: "What does a platypus look like? Generating customized prompts for zero-shot image classification"
_ICLR.cc/2023/Conference — Submitted to ICLR 2023_

### Official Review · Reviewer_LiMR · 2022-10-23

**Confidence:** 4
**Correctness:** 4
**Technical Novelty And Significance:** 2
**Empirical Novelty And Significance:** 4
**Recommendation:** 8

**Clarity, Quality, Novelty And Reproducibility:**

The paper is well-written and easy to follow. I think using GPT for generating prompts for CLIP style models is unique. The experiments are quite exhaustive demonstrating the effectiveness of the approach. The authors do a good job discussing each detail, and the work seems easy to reproduce.

**Strength And Weaknesses:**

Strengths
1. The approach is simple and effective. They show that by using the knowledge baked into LLMs like GPT can be used to improve zero-shot classification accuracy. I think this contribution is neat!

2. While some analysis is still missing in the paper (See weaknesses), I think the authors did a thorough empirical evaluation of their approach. I appreciated the author's effort on showing how different model sizes, number of LLM prompts, number of image prompts, LLM temperature affect performance. Per class accuracy difference (Figure 7) was also quite informative to look at!

Weaknesses:
1. Higher accuracy yes, but it doesn't remove the need for hand-designed prompts. As the authors mention in Figure 7, CuPL leads to significant accuracy boosts for certain classes (like mushrooms), but doesn't do well in other classes (canoe). This kinda means that hand-designed prompts can't be completely replaced because they bake in domain knowledge about the dataset.

2. I am curious how the GPT prompts are helping improve classification accuracy. It would be nice to sort of look at which words in the image prompt (maybe by looking at attention distribution) are important. Doing this both for standard as well as LLM based image prompts might help understand why there is such a drastic jump ( +/- ~40%) in accuracy for different classes.
    - I found the 40% accuracy jump for `mushroom' quite surprising. My first intuition was that the proposed approach will work better for classes that have distinct parts (cars, scooters, chair, etc). I wonder why LLM based prompt helped for a category like a mushroom?

3. Diversity Analysis (Figure 5) done on single LLM-prompt. It's quite possible that different LLM prompts, produces the same Image prompt. So an analysis of how different all the image prompts generated from different LLM prompts for a single image are would be nice.

4. In figure 6,
    - how did they choose which prompt to select when using fewer prompts. For instance, when only using 1 LLM prompt, which prompt did they use?
    - I also didn't understand the caption for Figure 6 (left). The caption says that CuPL outperforms the baseline with just 3 hand-written sentences. But in the figure, CuPL is outperforming the baseline even with 1 LLM prompt. Can the authors please clarify?

5. May I suggest adding a baseline that only uses 'a photo of a {}' prompt? The proposed approach lies somewhere in the spectrum of no-effort (using one standard prompt for everything), and high-effort (using standard hand-written prompts). We see where it lies with respect to high-effort. It'd be great to also see how the proposed approach compares to a _no-effort_ baseline.

**Summary Of The Paper:**

This work proposes to replace hand-written prompts used in CLIP style models with GPT-generated prompts for zero-shot image classification. The authors show that their approach (CuPL) of using LLMs to generate diverse text prompts, can improve classification accuracy while removing the need for hand-crafted prompts.

**Summary Of The Review:**

Overall, I like the paper. The paper presents a simple contribution and shows its effectiveness through thorough empirical evaluation. However, some analysis is still missing from the paper to build a better understanding of why the proposed approach is working. After reading the paper, apart from empirical evidence, there is little else to see the benefits of using the proposed approach. I think providing some of the analysis mentioned in my review will make the paper stronger and more insightful.

Update after rebuttal: I thank the authors to provide answers to my questions and incorporating all feedback! I think the paper presents a simple to understand/implement/use contribution that is very effective! I quite like the paper and I am bumping my rating to 8 to reflect that.

---

> ### Author Response · Authors · 2022-11-17
> **Thank you for your suggestions + Additional Experiments**
>
> Thank you for your helpful review, we are glad that you liked the paper and found our method unique. We have addressed a number of concerns and added **additional experiments**, especially around the analysis of CuPL prompts. We address each of your points one by one below:
>
>
> > Higher accuracy yes, but it doesn't remove the need for hand-designed prompts. As the authors mention in Figure 7, CuPL leads to significant accuracy boosts for certain classes (like mushrooms), but doesn't do well in other classes (canoe). This kinda means that hand-designed prompts can't be completely replaced because they bake in domain knowledge about the dataset.
>
> While our method does not completely eliminate the need for hand designed prompts, we believe that they do bake in significantly less domain knowledge. All of our LLM-prompts are very general and constructed without knowledge of the data domain (e.g. “What does a {} look like?”). We believe that it is a useful contribution to decrease (while not eliminate) the number of hand written prompts and the data knowledge in those prompts.
>
>
>
> > I am curious how the GPT prompts are helping improve classification accuracy… I found the 40% accuracy jump for `mushroom' quite surprising.
>
>
> Thank you for this suggestion regarding analysis of CuPL prompts. We have added a number of experiments to further explore this direction. In **Appendix F**, we **analyze** CuPL improvement in the **image embedding space**, and break down accuracy by **co-occurance of other ImageNet classes**. Additionally, in **Appendix H**, we now provide an error analysis, and find that **CuPL is more likely to make errors that are visually similar to the correct label, while the baseline makes errors that are linguistically similar**. While not exhaustive, we hope this provides an initial step in understanding the attributes of a high accuracy prompt for open vocabulary models
>
>
>
> > Diversity Analysis (Figure 5) done on single LLM-prompt. It's quite possible that different LLM prompts, produces the same Image prompt. So an analysis of how different all the image prompts generated from different LLM prompts for a single image are would be nice.
>
> Thank you for this suggestion! We agree this is valuable analysis and have added it to **Appendix I**.
>
>
> > In figure 6, how did they choose which prompt to select when using fewer prompts. For instance, when only using 1 LLM prompt, which prompt did they use?
>
> We selected this in a greedy manner, at each step 1-5 adding in the Image-prompts from the LLM-prompt that lead to the highest accuracy gain. We will add this clarification.
>
>
> > I also didn't understand the caption for Figure 6 (left). The caption says that CuPL outperforms the baseline with just 3 hand-written sentences. But in the figure, CuPL is outperforming the baseline even with 1 LLM prompt. Can the authors please clarify?
>
> Thank you very much for pointing this out. This was a mistake, it should say 1 hand-written sentence. This has been corrected
>
>
> > May I suggest adding a baseline that only uses 'a photo of a {}' prompt?
>
> Thank you for this suggestion! We have added this comparison to **Appendix E**. We find that **CuPL outperforms this baseline** both with one hand-written sentence and with one generated prompt.

---

### Official Review · Reviewer_DGAE · 2022-10-24

**Confidence:** 4
**Correctness:** 4
**Technical Novelty And Significance:** 2
**Empirical Novelty And Significance:** 2
**Recommendation:** 3

**Clarity, Quality, Novelty And Reproducibility:**

This paper is written well and easy to understand because the method is rather simple. It might not be easy to reproduce the results because this requires users to query GPT-3 many times, which is not free.

**Strength And Weaknesses:**

Strength

+It is interesting to generate prompts from LLM. Querying the description of an object class from LLM makes a lot sense because LLM is learned from large text corpus, covering knowledge about most of vocabularies.

Weakness

-I found this paper has little technical novelty. The proposed method generates prompts for each class from the outputs of a LLM i.e., GPT-3 and then use those prompts for zero-shot inference with the pretrained CLIP, which is straightforward and expected to achieve some improvement.

-The author's main contribution is querying GPT but the design decision of why a particular prompt is chosen to query GPT3 for CuPL base and full are not discussed or ablated.

-To my knowledge, using GPT-3 is not entirely free, which makes the method less appealing. Does this approach work well with other free LLM?

-In Fig. 5, increasing the temperature of LLM also increases the randomness of the outputs. What is the standard deviation at each temperature?

-The performance improvement are very minor and comes with a decrease in performance on other similar classes as shown in figure 7. This insight indicates that such a simple query setup is suboptimal for finegrained setting. The authors have not tried to mitigate this basic drawback.

-Detailed analysis on what makes the GPT3 prompts work better than human designed ones is missing when it is the main contribution of the work. For example in figure 5 we see that GPT3 prompts only surpass generic hand designed prompts at ~0.56 temperature. Why is this so? How do the outputs of the language model change with this temperature setting and what contributes to this change. If the method is this sensitive to temperature value, it adds to additional labelling cost of running the GPT3 model to ablate over the temperature value. How sensitive is this temperature value on different datasets? This needs to be analysed and discussed in light of labelling cost of generic human prompts.



**Summary Of The Paper:**

This paper proposes to generate prompts for zero-shot learning inference with CLIP by querying the large language model like GPT. The results show that the class prompts generated from GPT improves the zero-shot classification performance across many datasets.

**Summary Of The Review:**

Due to the weakness mentioned above, I think this paper does not have sufficient contributions for ICLR. I would recommend a reject and suggest the authors submit it to a workshop.

---

> ### Author Response · Authors · 2022-11-17
> **Thank you! Additional Experiments + Open source LLM**
>
> Thank you very much for your review of this work. We address each of your concerns point-by-point below
>
> > I found this paper has little technical novelty.
>
> Thank you for the constructive feedback. We are glad that you think our method is “straightforward and expected to achieve some improvement” — we thought the same! That is why we conducted extensive experiments to prove that. We believe that even “straightforward” and “expected” hypotheses need to be validated scientifically, or perhaps more so, as they are often easily taken for granted without careful analyses. Science is a process of furthering our understanding of the world by both validating (strengthening) and invalidating (updating) our pre-existing priors. Our work happens to fall into the first category, which we believe should be regarded as equally important as the second.
>
>
> > To my knowledge, using GPT-3 is not entirely free, which makes the method less appealing. Does this approach work well with other free LLM?
>
>
> Thank you for this suggestion! We agree it is a valuable direction to show performance of open-source models until large high performance models become open source. We have added this experiment to **Appendix D**. We find that **CuPL still outperforms human written prompts with the open source GPT-J-6B model**, although this does not match the performance of the GPT-3 model.
>
>
> > The performance improvement are very minor and comes with a decrease in performance on other similar classes as shown in figure 7. This insight indicates that such a simple query setup is suboptimal for finegrained setting. The authors have not tried to mitigate this basic drawback.
>
> While there is a decrease in accuracy for classes which are very similar to another class, we believe CuPL can still have high accuracy in fine-grained settings, as we show a performance increase on a number of fine-grained datasets, including OxfordPets and FGVC aircrafts.
>
>
> > Detailed analysis on what makes the GPT3 prompts work better than human designed ones is missing when it is the main contribution of the work.
>
> Thank you for this suggestion, and we have added a number of experiments to address this. In **Appendix G**, we **analyze** CuPL improvement in the **image embedding space**, and break down accuracy by **co-occurrence of other ImageNet classes**. Additionally, in **Appendix H**, we now provide an error analysis, and find that **CuPL is more likely to make errors that are visually similar to the correct label, while the baseline makes errors that are linguistically similar**. While not exhaustive, we hope this provides an initial step in understanding the attributes of a high accuracy prompt for open vocabulary models

---

### Official Review · Reviewer_ECmW · 2022-10-26

**Confidence:** 5
**Correctness:** 3
**Technical Novelty And Significance:** 1
**Empirical Novelty And Significance:** 2
**Recommendation:** 3

**Clarity, Quality, Novelty And Reproducibility:**

The paper is well written with clear presentation of the technical details, figures and tables. However, as discussed above, the novelty of the work is very limited. Also, how the proposed method can be applied for classification on new datasets without significant manual engineering and access to GPT3 is not clear.

**Strength And Weaknesses:**

**Strengths**:

- The paper is well-written and easy to follow.
- The idea is very simple and performs reasonably well on standard classification datasets.

**Weaknesses**:

- The technical novelty of the paper is very limited. It simply combines GPT3 with CLIP without a clear motivation. It is not surprising that output of a large language model will have better descriptions/knowledge compared to vanilla prompt engineering like "a photo of a". So, what is the main contribution of the paper need to be clearly discussed in the paper. Why someone will adopt this approach given that it often requires higher computational resources for inferencing a GPT3 model (and sometime not accessible for all)?

- Although the proposed framework uses a strong LLM like GPT3, the zero-shot classification performance on downstream datasets are very minimal. Training a continuous prompt, e.g., CoOp in Learning to Prompt for Vision-Language Models using 1-2 labeled samples performs significantly better than hand engineered prompts. It is often very practical to have 1-2 labeled samples per class inn many applications.  It is not clear why learning a single prompt using few labeled samples is not practical but the use of GPT3 can be considered practical for zero-shot classification? A thorough comparison and discussion with other prompt learning methods should be included in the paper.

- How does the proposed framework with only one prompt compare to the baseline that uses prompt like "a photo of a" (no ensemble of prompts)? This is a very important experiment to verify the effectiveness of the proposed framework in fair manner.

- What other generative LLMs besides GPT3 can be used and how are they comparable to the current one?

- The experiments only show performance on CLIP, while there exists many other recent vision-langauge models, e.g., DeCLIP, FILIP, CLOOB, CyCLIP etc. Authors should perform experiments on a variety of V&L models to demonstrate the generalizability of the proposed method for generating better prompts.

- Like original CLIP, CuPL still requires handcrafted prompts for LLM which is suboptimal. Identifying the right hand-crafted prompt is a non-trivial task, which often requires significant amount of time and domain-specific heuristics.

**Summary Of The Paper:**

This paper presents combines CLIP with large language models like GPT3 to create customized prompts for zero-shot image classification. Specifically authors leverage the knowledge contained in LLMs in order to generate many descriptive sentences that are customized for each object category for prompting CLIP models. Experiments on 15 downstream classification datasets show the effectiveness of the proposed method over handcrafted prompts that were used in the original CLIP adaptation.

**Summary Of The Review:**

I’d like to rate the current submission as a clear rejection due to very limited technical novelty and lack of convincing experiments. Despite all the changes, I still feel the paper will not have enough contributions to be accepted for ICLR.

---

> ### Author Response · Authors · 2022-11-17
> **Thank you, Response + Additional Experiments**
>
> Thank you for your valuable review of our work. Below we address each of your comments individually.
>
> > “Why someone will adopt this approach given that it often requires higher computational resources for inferencing a GPT3 model (and sometime not accessible for all)?”
>
> We believe our method offers several benefits over traditional hand-crafted prompts. First, while hand writing prompts may not have such a clear cost attached, time and effort for curating 80 hand-written prompts by humans is costly.
>
> Additionally, the cost associated with generating CuPL prompts is not very high when compared to computational costs in machine learning research. For example, generating 10 sentences, each >50 tokens for all 1000 ImageNet class is at most 50 USD, using the most expensive current model. Furthermore, this is a one time cost. Finally, should new classes be added, they can be added incrementally at 0.05 USD a class without requiring new computation for the existing 1000 classes, while many models would have to be completely fine-tuned for any changes. While we agree there is some additional computation involved, we argue that this is true of many machine learning advances and the cost of our method is low compared to the benefits.
>
> We have also since added evaluations of our method on **open-sourced LLMs like GPT-J (Appendix D)**, which alleviates the inference cost. We find that the improvements on GPT-3 transfer to other LLMs.
>
>
>
>
> > Training a continuous prompt, e.g., CoOp in Learning to Prompt for Vision-Language Models using 1-2 labeled samples performs significantly better than hand engineered prompts. It is often very practical to have 1-2 labeled samples per class inn many applications. It is not clear why learning a single prompt using few labeled samples is not practical but the use of GPT3 can be considered practical for zero-shot classification?
>
> We agree that CoOp is an excellent method under certain circumstances. However we also believe that there are many situations in which it is valuable to be zero-shot rather than few shot.
>
> Firstly, one of the large benefits of open-vocabulary is that the classes can be selected ad hoc at inference time, without additional training. With few shot this is no longer the case, and adding more/different classes means more training. This takes away much of the flexibility of models like CLIP. With our method however, no additional training is required. CuPL prompts can be generated and added to the potential classes in the open vocab model all at inference.
>
> Additionally, while a few samples is feasible in many cases, this is not true in all cases (for example in scientific or medical cases). Also, few shot becomes less practical when there are many thousands or millions of classes, leading to the need to store and train on many images.
>
> We agree there are many cases, when zero-shot is not the right setting, but we also believe this is a useful field of study for the cases when training is not convenient.
>
>
> > How does the proposed framework with only one prompt compare to the baseline that uses prompt like "a photo of a" (no ensemble of prompts)?
>
> Thank you for this suggestion! We have added this comparison to **Appendix E**. We find that **CuPL outperforms this baseline** both with one hand-written sentence and with one generated prompt.
>
>
> > What other generative LLMs besides GPT3 can be used and how are they comparable to the current one?
>
> Thank you for this suggestion, and we agree that it is valuable to add experiments with open sourced generative LLMs. We have added more experiments to **Appendix D**. We find that **CuPL still outperforms human written prompts with the open source GPT-J-6B model,** although GPT-3, being a more powerful model, still offers the best performance.
>
>
>
>
> > The experiments only show performance on CLIP, while there exists many other recent vision-langauge models, e.g., DeCLIP, FILIP, CLOOB, CyCLIP
>
> Thank you for pointing out other vision-language models we could try CuPL with. We think that the generality of CuPL should suggest that the improvements are independent of model choices. While in our paper, we fixed the open vocabulary model CLIP and varied the LLM (different sizes of GPT-3, and the newly added GPT-J in Appendix D), a natural expansion of this work would be swapping CLIP with other equivalent models. We hope to cover this in our future work.
>
>
> > Like original CLIP, CuPL still requires handcrafted prompts for LLM which is suboptimal.
>
> We agree that it would be ideal to have no hand constructed prompts at all. However, we believe that bringing the number of overall hand written prompts down to 5 from 80 for imagenet is a large improvement. Additionally, we show 3 prompts which are effective for a wide number of datasets. Using these prompts for future datasets does decrease the amount of engineering for these future datasets down to 0.

---

> > ### Author Response · Authors · 2022-11-17
> > **Cont.**
> >
> > > Also, how the proposed method can be applied for classification on new datasets without significant manual engineering and access to GPT3 is not clear.
> >
> > On the contrary, we believe that **CuPL is a solution to the manual engineering** that is currently necessary to achieve high performance on new datasets with CLIP. Currently, new Image-prompts templates need to be constructed for each new dataset. However, with CuPL (base), we achieve higher performance on 13 out of 15 datasets using **the same 3 prompt templates for each dataset**. Therefore it is much easier than before to perform classification on a new dataset.
> >
> > Additionally, GPT3 has a publicly available API, that (while not totally free) is at most $50 to generate CuPL (full) prompts for ImageNet (the most expensive experiment in this work). Additionally, we now show that CuPL outperforms human written prompts with the **open source GPT-J-6B model, which we have added to Appendix D**.

---

> > > ### Comment · Reviewer_ECmW · 2022-12-01
> > > **Final Response**
> > >
> > > I thank the authors for their response and effort on the new experiments. After carefully reading the authors response including other reviewers concerns, I am still not convinced with the key technical contributions of the paper. Even with the new experiment on GPT-J, I still feel the paper does not have enough contributions to be accepted for ICLR (the current approach is just a simple combination of GPT with CLIP without a clear motivation). As I said before, it is often very practical to have one labeled sample per class which can be used for prompt learning like CoOp. The performance of the proposed method even with the use of very strong LLM like GPT3 does not outperform such a baseline which limits the usability of the method on many applications. Moreover, authors claim on 80 manual prompts is not always required for every target downstream tasks. While CLIP showed performance by ensembeling 80 prompts on ImageNet, only one prompt like "a photo of a" works reasonably well in many downstream applications. Finally, given very limited technical novelty, I would recommend authors to perform experiments on recent vision-language models, e.g., DeCLIP, FILIP, CLOOB, CyCLIP to verify the generalizability of the proposed method. Based on all these points, I am keeping my initial rating on this paper.

---

### Official Review · Reviewer_UY2k · 2022-10-27

**Confidence:** 4
**Correctness:** 4
**Technical Novelty And Significance:** 3
**Empirical Novelty And Significance:** 2
**Recommendation:** 6

**Clarity, Quality, Novelty And Reproducibility:**

Clarity questions:
How are “hyperparameters” like the number of prompts and which specific prompts to choose decided? Is there a “validation set”?
Is there qualitative or anecdotal analysis of how “diverse” the generated image prompts are after increasing the temperature? For example, would higher temperatures also “hurt” by introducing more incorrect knowledge?

Minor comments:
Figure 6: CuPL outperforms the baseline even with just three hand-written sentence. Should be one hand-written sentence.
Also, how many prompts are there for Figure 6 left figure? If it is 10, it means that 10 CuPL image prompts can outperform 80 standard image prompts which is less than 25 in the caption.

The paper is novel and easy to reproduce.

**Strength And Weaknesses:**

Strength:
- The paper is well written. The method is intuitive and the performance gain is good.
- The authors did different ablation studies to show how different design choices would affect performance including the number of prompts, diversity of prompts, model size, etc. These ablations are helpful for the audience to understand the full picture of this method.

Weaknesses:
- Although the general idea is good, the authors only consider one task in this paper which is zero-shot image classification. It is more interesting for tasks like open vocabulary segmentation/detection etc. This method can even be used for image caption retrieval where LLM is used to paraphrase the caption. It is more convincing that the proposed method can universally work on different tasks.
Even for imagenet, it is more interesting to see the performance on the full imagenet21k instead of the 1k set. In addition, original CLIP is evaluated on different relatives of Imagenet like imagenet-v2, imagenet-R etc. which would also be good to have in this paper.
- It would be better if there can be a clear analysis or explanation of why the standard prompts and CuPL prompts have different preferences as shown in figure 7. It is a really interesting phenomenon to me, and it may also reveal some hints on how to improve CuPL.

**Summary Of The Paper:**

Open vocabulary models require natural language prompts as an intermediate medium between category and visual content, thus it is important to choose a good prompt. In this paper, the authors study how to use a pretrained language model to help generate better language prompts for open vocabulary tasks. The authors feed the categories with language prompt templates to an LLM and use the output of the LLM as the image prompts. The authors show that the prompts generated by their method can achieve better performance on zero-shot image classification benchmarks.

**Summary Of The Review:**

The authors propose a novel method that is simple and has the potential to apply to many different tasks. However, the authors did not show that this method is widely applicable which makes the paper weaker. My decision is borderline accept.

---

> ### Author Response · Authors · 2022-11-17
> **Thank You + Additional Analysis**
>
> Thank you for your valuable feedback! We appreciate that you found our work novel and our paper well-written. To address some of your concerns, we have added additional experiments to the newest version of our work. We address each of your points below:
>
>
> > Although the general idea is good, the authors only consider one task in this paper which is zero-shot image classification. It is more interesting for tasks like open vocabulary segmentation/detection etc. This method can even be used for image caption retrieval where LLM is used to paraphrase the caption. .... In addition, original CLIP is evaluated on different relatives of Imagenet like imagenet-v2, imagenet-R etc. which would also be good to have in this paper.
>
> Thanks for your great insight. We agree that zero-shot image classification alone only demonstrates the tip of the iceberg in terms of the potential of our method. We hope the positive result we’ve shown on zero-shot image classification would encourage fellow researchers to apply this idea to segmentation, detection, retrieval, etc. Upon your suggestion, we have added evaluations on distribution shift using **ImageNet-V2** and **ImageNet-Sketch**, in **Appendix F**. We find that our method is robust to distribution shifts, and improves upon CLIP (standard).
>
> > Even for imagenet, it is more interesting to see the performance on the full imagenet21k instead of the 1k set
>
> We agree that it would be beneficial to increase the scale of our evaluation. Currently, we do not have the resource or budget to carry out experiments at 20x scale, however, as we open sourced our code and methodology, we are eager to invite the community to validate our findings across different tasks and scales.
>
>
> > It would be better if there can be a clear analysis or explanation of why the standard prompts and CuPL prompts have different preferences as shown in figure 7. It is a really interesting phenomenon to me, and it may also reveal some hints on how to improve CuPL
>
> Thank you for this suggestion, and we have added a number of experiments to address this. In **Appendix G**, we **analyze** CuPL improvement in the **image embedding space**, and break down accuracy by **co-occurrence of other ImageNet classes**. Additionally, in **Appendix H**, we now provide an error analysis, and find that **CuPL is more likely to make errors that are visually similar to the correct label, while the baseline makes errors that are linguistically similar**. While not exhaustive, we hope this provides an initial step in understanding the attributes of a high accuracy prompt for open vocabulary models
>
> > How are “hyperparameters” like the number of prompts and which specific prompts to choose decided? Is there a “validation set”?
>
> Thank you for your question. Number of prompts is a design choice — we didn’t have to tune it with any validation set.
>
> > Is there qualitative or anecdotal analysis of how “diverse” the generated image prompts are after increasing the temperature? For example, would higher temperatures also “hurt” by introducing more incorrect knowledge?
>
> This is a great question. We have since added **Appendix J: “Temperature Analysis.”**  The conclusion is higher temperature does increase the diversity of prompts, as expected.
>
> > Minor comments: Figure 6: CuPL outperforms the baseline even with just three hand-written sentence. Should be one hand-written sentence. Also, how many prompts are there for Figure 6 left figure? If it is 10, it means that 10 CuPL image prompts can outperform 80 standard image prompts which is less than 25 in the caption.
>
> Thank you for this correction, we have changed it to say one hand-written sentence. And you are correct that 10 CuPL image-prompts already outperforms 80, we have adjusted the phrasing in the figure 6 caption to be more precise.

---

### Author Response · Authors · 2022-11-17
**Thank you all reviewers! + Overview of updates**

We want to thank the reviewers for their careful read of our work and valuable suggestions. After consideration of their feedback, we have added a number of new results which we believe greatly strengthen our overall contribution.

**Open-Source LLM**: A number of reviewers have pointed out that our work would benefit from showing it is reproducible with an open source Language Model. We implement this with GPT-J-6B and present our findings in **Appendix D**. While CuPL with GPT-3 still produces the highest accuracy, we are able to show that **an open source model can still surpass human written prompts** with far less human written sentences overall.


**Single Sentence Baseline**: We add a baseline in **Appendix E** which uses only a single hand-constructed sentence, as multiple reviewers suggested. We find that **CuPL outperforms this baseline** both with one hand-written sentence and with one generated prompt.


**Robustness**: We show that our method remains **robust to distribution shift** by showing improvements on two common ImageNet distributions in **Appendix F**.


**Prompt Analysis**: We have added several additional pieces of analysis in **Appendix G** to better understand the effect of our prompts, as was requested by several reviewers. First, we analyze CuPL improvement in the **image embedding space** which allows us to visualize how proximity to related categories affects accuracy. Secondly, we break down accuracy by **co-occurrence of other ImageNet classes**.


**Error Analysis**: Additionally, we provide analysis on the type of errors made by the baseline compared to our method in **Appendix H**.  We find that **CuPL is more likely to make errors that are visually similar to the correct label, while the baseline makes errors that are linguistically similar**.


**LLM-prompt effect on Image-Prompts**: As requested by a reviewer, we provide analysis on the distribution ofImage- prompts based on the LLM-prompt that was used to generate them. We find significant overlap in the distribution of Image-prompts, even when generated with different LLM-prompts.This analysis is in Appendix I.


**Temperature Analysis**: Additionally, we provide analysis on the effect of temperature in generation. We show that higher temperatures lead to a wider distribution in the text embedding space. We also provide qualitative results of prompt generations at different temperatures. These results can be found in Appendix J and K.

We thank all reviewers again for their high quality feedback and experiment suggestions that allowed us to strengthen our work

---

### Decision · Program_Chairs · 2023-01-20

**Decision:**

Reject

**Justification For Why Not Higher Score:**

Two reviewers recommend rejection; one recommends acceptance but raises several concerns.

**Justification For Why Not Lower Score:**

N/A

**Metareview: Summary, Strengths And Weaknesses:**

The paper proposes the use of GPT-3 to generate a prompt for a zero-shot object classification method. Upon the request of the reviewers, the authors add further results with other language models. While the proposed method is simple and effective in some settings, the authors question whether the technical contribution is sufficient, whether the use of GPT-3 (which is not free) is warranted (and results with other LMs are not as good), and practicality (given that prompts can be learned with a small amount of data).